# CKConv: Continuous Kernel Convolution For Sequential Data

**David W. Romero**[1]**, Anna Kuzina**[1]**, Erik J. Bekkers**[2]**, Jakub M. Tomczak**[1]**, Mark Hoogendoorn**[1]
[1] Vrije Universiteit Amsterdam     [2] University of Amsterdam
The Netherlands
{d.w.romeroguzman, a.kuzina}@vu.nl

## Abstract

Conventional neural architectures for sequential data present important limitations. Recurrent neural networks suffer from exploding and vanishing gradients, small effective memory horizons, and must be trained sequentially. Convolutional neural networks cannot handle sequences of unknown size and their memory horizon must be defined a priori. In this work, we show that these problems can be solved by formulating the convolutional kernels of CNNs as continuous functions. The resulting *Continuous Kernel Convolution* (CKConv) handles arbitrarily long sequences in a parallel manner, within a single operation, and without relying on any form of recurrence. We show that *Continuous Kernel Convolutional Networks* (CK-CNNs) obtain state-of-the-art results in multiple datasets, e.g., permuted MNIST, and, thanks to their continuous nature, are able to handle non-uniformly sampled datasets and irregularly-sampled data natively. CKCNNs match or perform better than neural ODEs designed for these purposes in a faster and simpler manner.

## 1 Introduction

Recurrent Neural Networks (RNNs) have long governed tasks with sequential data (Rumelhart et al., 1985; Hochreiter & Schmidhuber, 1997; Chung et al., 2014). Their main ingredient are *recurrent units*: network components with a recurrence formulation which grants RNNs the ability to be unrolled for arbitrarily many steps and handle sequences of arbitrary size. In practice, however, the effective *memory horizon* of RNNs, i.e., the number of steps the network retains information from, has proven to be surprisingly small, most notably due to the *vanishing gradients problem* (Hochreiter, 1991; Bengio et al., 1994). Interstingly, it is the very recurrent nature of RNNs that allows them to be unrolled for arbitrarily many steps which is responsible for vanishing gradients (Pascanu et al., 2013b). This, in turn, hinders learning from the far past and induces a small effective memory horizon.

Convolutional Neural Networks (CNNs) (LeCun et al., 1998) have proven a strong alternative to recurrent architectures as long as relevant input dependencies fall within their memory horizon, e.g., Conneau et al. (2016); Oord et al. (2016); Dai et al. (2017); Dauphin et al. (2017); Bai et al. (2018a). CNNs avoid the training instability and vanishing / exploding gradients characteristic of RNNs by avoiding *back-propagation through time* (Werbos, 1990) altogether. However, these architectures model convolutional kernels as a sequence of independent weights. As a result, their memory horizon must be defined *a-priori*, and larger memory horizons induce a proportional growth of the model size.

In this work, we provide a solution to these limitations. We propose to view a convolutional kernel as a continuous function parameterized by a small neural network instead of a sequence of independent weights. The resulting *Continuous Kernel Convolution* (CKConv) enjoys the following properties:

- CKConvs can define arbitrarily large memory horizons within a single operation. Consequently, *Continuous Kernel Convolutional Neural Networks* (CKCNNs) detach their memory horizon from *(i)* the depth of the network, *(ii)* the dilation factors used, and *(iii)* the size of the network.

- CKConvs do not rely on any form of recurrence. As a result, CKCNNs *(i)* can be trained in parallel, and *(ii)* do not suffer from vanishing / exploding gradients or small effective memory horizons.

- Continuous convolutional kernels can be evaluated at arbitrary positions. Consequently, CKConvs and CKCNNs can be readily used on irregularly sampled data, and data at different resolutions.

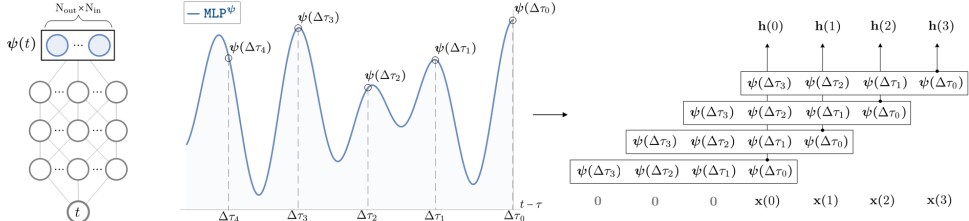

Figure 1: Continuous Kernel Convolution (CKConv). CKConv views a convolutional kernel as a vector-valued continuous function $\psi : \mathbb{R} \to \mathbb{R}^{N_{out} \times N_{in}}$ parameterized by a small neural network $\text{MLP}^{\psi}$. $\text{MLP}^{\psi}$ receives a time-step and outputs the value of the convolutional kernel at that position. We sample convolutional kernels by passing a set of relative positions $\{\Delta \tau_i\}$ to $\text{MLP}^{\psi}$, and perform convolution with the sampled kernel next. Since $\text{MLP}^{\psi}$ is a continuous function, CKConvs can (*i*) construct arbitrarily large kernels, (*ii*) generate kernels at different resolutions, and (*iii*) handle irregular data.

We observe that continuous kernel parameterizations previously used to handle irregular data *locally*, e.g., Schütt et al. (2017); Wu et al. (2019), are not adequate to model long-term dependencies. This is due to the inability of their kernels to model long spatial complex functions (Sec. 4.2). Contrarily, CKConvs perfectly describe long complex non-linear, non-smooth functions by parameterizing their kernels as SIRENs (Sitzmann et al., 2020): implicit neural representations with $\mathrm{Sine}$ nonlinearities. Shallow CKCNNs match or outperform state-of-the-art approaches on several tasks comprising stress tests, continuous, discrete and irregular data, as well as resolution changes. To the best of our knowledge, we are first to observe the potential of continuous convolutional kernels to model long-term dependencies, and to provide an useful parameterization to this end.

## 2 Related Work

**Continuous kernel formulation.** Continuous formulations for convolutional kernels were introduced to handle irregularly sampled 3D data *locally* (Schütt et al., 2017; Simonovsky & Komodakis, 2017; Wang et al., 2018; Wu et al., 2019). As discrete convolutions learn independent weights for specific relative positions, they cannot handle irregularly sampled data effectively. Following work focuses on point-cloud applications (Fuchs et al., 2020; Hu et al., 2020; Shi et al., 2019; Thomas et al., 2018). Other approaches include Monte Carlo approximations of continuous operations (Finzi et al., 2020). Our work proposes a new broad flavor of applications for which continuous kernels are advantageous.

**Implicit neural representations.** Implicit neural representations construct continuous data representations by encoding the input in the weights of a neural network (Mescheder et al., 2019; Park et al., 2019; Sitzmann et al., 2020). This leads to numerous advantages over conventional (discrete) data representations, e.g., memory efficiency, analytic differentiability, with interesting properties for several applications, e.g., generative modelling (Dupont et al., 2021; Schwarz et al., 2020).

Since we model convolutional kernels as continuous functions and parameterize them via neural networks, our approach can be understood as *implicitly representing the convolutional kernels of a conventional CNN*. Different is the fact that these convolutional kernels are not known a-priori, but learned as a part of the optimization task of the CNN. Making the connection between implicit neural representations and continuous kernel formulations explicitly brings substantial insights for the construction of these kernels. In particular, it motivates the use of $\mathrm{Sine}$ nonlinearities (Sitzmann et al., 2020) to parameterize them, which leads to significant improvements over the $\mathrm{ReLU}$, $\mathrm{LeakyReLU}$, and $\mathrm{Swish}$ nonlinearities used so far for this purpose (Sec. 4.2).

## 3 The Convolution and Common Kernel Parameterizations

**Notation.** $[n]$ denotes the set $\{0, 1, \dots, n\}$. Bold capital and lowercase letters depict vectors and matrices, e.g., $\mathbf{x}$, $\mathbf{W}$, sub-indices index vectors, e.g., $\mathbf{x} = \{x_c\}_{c=1}^{N_{in}}$, parentheses index time, e.g., $\mathbf{x}(\tau)$ is the value of $\mathbf{x}$ at time-step $\tau$, and calligraphic letters depict sequences, e.g., $\mathcal{X} = \{\mathbf{x}(\tau)\}_{\tau=0}^{N_X}$.

**Centered and causal convolutions.** Let $\mathbf{x} : \mathbb{R} \to \mathbb{R}^{N_{in}}$ and $\boldsymbol{\psi} : \mathbb{R} \to \mathbb{R}^{N_{in}}$ be a vector valued signal and kernel on $\mathbb{R}$, such that $\mathbf{x} = \{x_c\}_{c=1}^{N_{in}}$ and $\boldsymbol{\psi} = \{\psi_c\}_{c=1}^{N_{in}}$. The convolution is defined as:

$$(\mathbf{x} * \boldsymbol{\psi})(t) = \sum_{c=1}^{N_{in}} \int_{\mathbb{R}} x_c(\tau) \psi_c(t - \tau) \, d\tau. \tag{1}$$

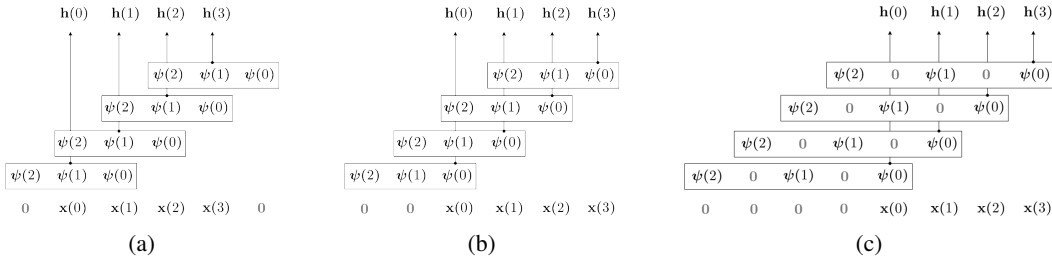

Figure 2: Discrete centered, causal, and dilated causal convolutions.

In practice, the input signal $\mathbf{x}$ is gathered via some sampling procedure. Resultantly, the convolution is effectively performed between the sampled input signal described as a sequence of finite length $\mathcal{X} = \{\mathbf{x}(\tau)\}_{\tau=0}^{N_X}$ and a convolutional kernel $\mathcal{K} = \{\boldsymbol{\psi}(\tau)\}_{\tau=0}^{N_X}$ described the same way:

$$(\mathbf{x} * \boldsymbol{\psi})(t) = \sum_{c=1}^{N_{in}} \sum_{\tau=-N_X/2}^{N_X/2} x_c(\tau)\psi_c(t-\tau). \tag{2}$$

Values $\mathbf{x}(\tau)$ falling outside of $\mathcal{X}$ are *padded* by a constant value often defined as zero (Fig. 2a).

The convolutional kernel is commonly *centered* around the point of calculation $t$. For sequence modeling this can be undesirable as future input values $\{\mathbf{x}(t-\tau)\}_{\tau=-N_X/2}^{-1}$ are considered during the operation. This is solved by providing a *causal formulation to the convolution*: a formulation in which the convolution at time-step $t$ only depends on input values at time-steps $(t-\tau) \leq t$ (Fig. 2b):

$$(\mathbf{x} * \boldsymbol{\psi})(t) = \sum_{c=1}^{N_{in}} \sum_{\tau=0}^{t} x_c(\tau)\psi_c(t-\tau). \tag{3}$$

In practice, causal convolutions are easily implemented via asymmetrical padding. In this work, we consider causal convolutions as default. Nevertheless, our analyses are also valid for centered ones.

**Discrete convolutional kernels.** By a large margin, most convolutional kernels $\boldsymbol{\psi}$ in literature are parameterized as a finite sequence of $N_K + 1$ independent learnable weights $\mathcal{K} = \{\boldsymbol{\psi}(\tau)\}_{\tau=0}^{N_K}$ (Fig. 2). As these weights are independent of one another, $N_K$ must be kept small to keep the parameter count of the model tractable. Hence, the kernel size is often much smaller than the input length: $N_K \ll N_X$. This parameterization presents important limitations:

- The memory horizon $N_K$ must be defined a priori.

- Since $N_K \ll N_X$, this parameterization implicitly assumes that the convolution $(\mathbf{x} * \boldsymbol{\psi})$ at position $t$ *only depends on input values at positions up to $\tau = N_K$ steps in the past*. Consequently, no functions depending on inputs $\mathbf{x}(t-\tau)$ for $\tau > N_K$ can be modeled.

- The most general selection of $N_K$ is given by a *global memory horizon*: $N_K = N_X$. Unfortunately, as discrete convolutional kernels are modeled as a sequence of independent weights, this incurs an extreme growth of the model size and rapidly becomes statistically unfeasible.

**Dilated convolutional kernels.** To alleviate these limitations, previous works propose to interleave kernel weights with zeros in order to cover larger memory horizons without additional weights (Fig. 2c). This formulation alleviates some of the previous limitations, but introduces additional ones:

- Dilated kernels are unable to model dependencies of input values falling in the interleaved regions.

- Several authors use dilated convolutions with varying dilation factors as a function of depth, e.g., (Bai et al., 2018a; Dai et al., 2017; Oord et al., 2016; Romero et al., 2020). By carefully selecting layer-wise dilation factors, one can assure that some kernel hits each input within the memory horizon of the network. However, due to the extreme sparsity of the formulation, it is difficult to estimate the effective amount of processing applied to the input. In addition, this layout ties together (*i*) the memory horizon, (*ii*) the depth, and (*iii*) the layer-wise dilation factors of the network, which effectively constraints the flexibility of the neural architecture design.

In contrast to the (dilated) discrete convolutions presented in this section, our proposed formulation allows handling arbitrarily long sequences with arbitrarily large, dense memory horizons in a single layer and under a fixed parameter budget.

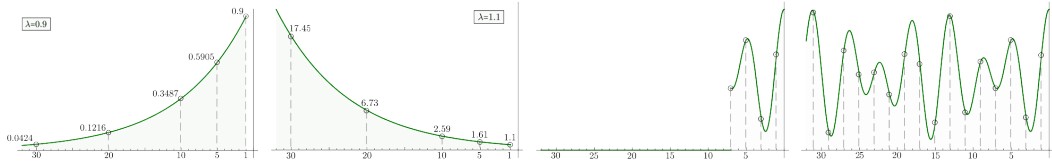

(a) Recurrent unit ($\lambda{\leq}1$)   (b) Recurrent unit (($\lambda{\geq}1$)   (c) Discrete kernel   (d) Continuous kernel

Figure 3: Functional family of recurrent units, discrete convolutions and CKConvs. For max. eigenvalues of $\mathbf{W}$, $\lambda{\neq}1$, recurrent units are restricted to exponentially decreasing ($\lambda{\leq}1$) or increasing ($\lambda{\geq}1$) functions (Figs. 3a, 3b). Discrete convolutions can describe arbitrary functions within their memory horizon but are zero otherwise (Fig. 3c). Conversely, CKConvs define arbitrary long memory horizons, and thus are able to describe arbitrary functions upon the entire input sequence (Fig. 3d).

## 4   CONTINUOUS KERNEL CONVOLUTION

In this section, we introduce our approach. First, we define it formally, analyze its properties, illustrate its connection to recurrent units, and elaborate on the functional family they can describe. Next, we discuss concrete parameterizations of continuous convolutional kernels, illustrate their connection to implicit neural representations, and show that our final kernels are able to fit complex functions.

### 4.1   FORMULATION AND PROPERTIES

**Arbitrarily large convolutional kernels.** We formulate the convolutional kernel $\psi$ as a continuous vector-valued function parameterized by a small neural network $\texttt{MLP}^{\psi}: \mathbb{R} \to \mathbb{R}^{N_{\text{out}} \times N_{\text{in}}}$ (Fig. 1, left). $\texttt{MLP}^{\psi}$ receives a relative position $(t{-}\tau)$ and outputs the value of the convolutional kernel at that position $\psi(t{-}\tau)$. As a result, an arbitrarily large convolutional kernel $\mathcal{K}{=}\{\psi(t{-}\tau)\}_{\tau=0}^{N_{\text{K}}}$ can be constructed by providing an equally large sequence of relative positions $\{t{-}\tau\}_{\tau=0}^{N_{\text{K}}}$ to $\texttt{MLP}^{\psi}$. For $N_{\text{K}}{=}N_{\text{X}}$, the size of the resulting kernel is equal to that of the input sequence $\mathcal{X}$, and thus it is able to model (global) long-term dependencies. The *Continuous Kernel Convolution* (CKConv) is given by:

$$(\mathbf{x} * \boldsymbol{\psi})(t) = \sum_{c=1}^{N_{\text{in}}} \sum_{\tau=0}^{t} x_c(\tau) \texttt{MLP}_c^{\psi}(t - \tau). \tag{4}$$

**Irregularly sampled data.** CKConvs are able to handle irregularly-sampled and partially observed data. To this end, it is sufficient to sample $\texttt{MLP}^{\psi}$ at positions for which the input signal is known and perform the convolution operation with the sampled kernel. For very non-uniformly sampled inputs, an inverse density function over the samples can be incorporated in order to provide an unbiased estimation of the convolution response (see Appx. A.1, Wu et al. (2019) for details).

**Data at different resolutions.** CKConvs can also process data at different resolutions. Consider the convolution $(\mathbf{x} * \boldsymbol{\psi})_{\text{sr}_1}$ between an input signal $\mathbf{x}$ and a continuous convolutional kernel $\psi$ sampled at a sampling rate $\text{sr}_1$. Now, if the convolution receives the same input signal sampled at a different sampling rate $\text{sr}_2$, it is sufficient to sample the convolutional kernel at the sampling rate $\text{sr}_2$ in order to perform an "equivalent" operation: $(\mathbf{x} * \boldsymbol{\psi})_{\text{sr}_2}$. As shown in Appx. A.2, it holds that:

$$(\mathbf{x} * \boldsymbol{\psi})_{\text{sr}_2}(t) \approx \frac{\text{sr}_2}{\text{sr}_1} (\mathbf{x} * \boldsymbol{\psi})_{\text{sr}_1}(t). \tag{5}$$

That is, convolutions calculated at different resolutions $\text{sr}_1$ and $\text{sr}_2$ are approximately equal up to a factor given by the resolution change. As a result, CKCNNs *(i)* can be trained in datasets with data at varying resolutions, and *(ii)* can be deployed at resolutions other than those seen during training.

We note that the previous features are hardly attainable by regular architectures, with an exception being RNNs with continuous-time interpretations, e.g., Gu et al. (2020a); Kidger et al. (2020).

**(Linear) recurrent units are continuous kernel convolutions.** Consider a recurrent unit of the form:

$$\mathbf{h}(\tau) = \sigma\big(\mathbf{W}\mathbf{h}(\tau - 1) + \mathbf{U}\mathbf{x}(\tau)\big) \tag{6}$$

$$\tilde{\mathbf{y}}(\tau) = \text{softmax}(\mathbf{V}\mathbf{h}(\tau)), \tag{7}$$

where $\mathbf{U}, \mathbf{W}, \mathbf{V}$ depict the *input-to-hidden*, *hidden-to-hidden* and *hidden-to-output* connections of the unit, $\mathbf{h}(\tau), \tilde{\mathbf{y}}(\tau)$ the hidden representation and the output at time-step $\tau$, and $\sigma$ a pointwise nonlinearity. As shown in Appx. A.3, we can express the hidden representation $\mathbf{h}$ of a linear recurrent unit, i.e., with $\sigma{=}\text{Id}$, as a convolution between the input $\mathbf{x}$ and a convolutional kernel $\psi(\tau){=}\mathbf{W}^{\tau}\mathbf{U}$ of size equal to the input. That is, as a continuous kernel convolution with an exponentially increasing

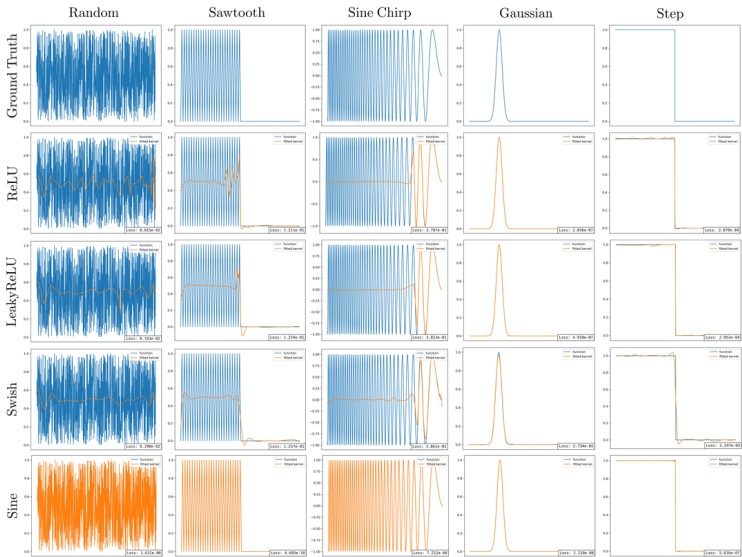

Figure 4: Approximation quality of MLPs with ReLU, LeakyReLU, Swish, and Sine nonlinearities. Networks with (smooth) piece-wise nonlinearities are unable to approximate non-smooth, non-linear functions. Sine networks, on the other hand, quickly approximate all target functions to near perfection. All networks share the same structure and vary only in the nonlinearity used.

or decreasing kernel (Fig. 3). Different authors show that nonlinear recurrent units are also restricted to the same functional family (Pascanu et al., 2013b; Arjovsky et al., 2016; Zhao et al., 2020).

**The functional family of continuous kernel convolutions.** From the previous observation, we can conclude that CKConvs are not only more general than discrete convolutions, but that the functional family they describe is also more general than that of (linear) recurrent units (Fig. 3).

### 4.2 THE CONTINUOUS CONVOLUTIONAL KERNEL MLP$^{\psi}$

**Convolutional kernels as point-wise MLPs.** Let $\{\Delta\tau_i{=}(t{-}\tau_i)\}_{i=0}^{N}$ be a sequence of relative positions. The continuous vector-valued convolutional kernel $\psi : \mathbb{R} \to \mathbb{R}^{N_{out} \times N_{in}}$ is parameterized as a neural network MLP$^{\psi}$ which maps each relative position $\Delta\tau_i$ to the value of the convolutional kernel at that position (Fig. 1, left). We refer to the nonlinearity used in MLP$^{\psi}$ as $\sigma$.

**What kind of kernels can MLP$^{\psi}$ generate?** Our method relies on the assumption that the neural network MLP$^{\psi}$ is able to model complex dependencies densely among all elements within the memory horizon. That is, it assumes that MLP$^{\psi}$ is able to generate arbitrary convolutional kernels.

To test this hypothesis, we fit existing MLP parameterizations, i.e., with ReLU, LeakyReLU and Swish nonlinearities, to long target functions of varying level of smoothness and non-linearity (Fig. 5). We observe that existing parameterizations can approximate simple functions, e.g., Gaussian, step functions, but for increasing levels of non-linearity and non-smoothness, they fail by a large margin. For our analysis, this means that CKConvs with ReLU, LeakyReLU and Swish parameterizations are *not* able to represent complex input dependencies. In our ablation studies (Appx. D) we verify that CKCNNs with these kernels consistently perform worse than our proposed parameterization.

**Convolutional kernels as implicit neural representations.** We notice that parameterizing a convolutional kernel with a neural network is equivalent to constructing an implicit neural representation of the kernel, with the subtle difference that our target objective is not known a-priori, but learned as part of the optimization task of the CNN. Implicit neural representations study generic ways to represent data living in low-dimensional spaces, e.g., $\mathbb{R}^2$, via neural networks, and thus, despite this difference, constitute an interesting starting point for the parameterization of continuous convolutional kernels. In particular, recent works noticed that neural networks with piece-wise activation functions are unable to model high-frequencies. To alleviate this limitation, they introduce random Fourier features (Tancik et al., 2020) and Sine nonlinearities (Sitzmann et al., 2020).

Based on these observations, we repeat the fitting experiment for a SIREN (Sitzmann et al., 2020): a MLP with hidden layers of the form $\mathbf{y} = \text{Sine}(\omega_0[\mathbf{W}\mathbf{x} + \mathbf{b}])$. That is with Sine nonlinearities,

and a non-learnable value $\omega_0$ that serves as a prior for the oscillations of the output. We observe that a SIREN quickly approximates *all* target functions to near perfection regardless of their grade of smoothness or nonlinearity. Even a sequence of random noise. This implies that, contrary to other parameterizations, CKConvs with SIREN kernels have the ability to model complex input dependencies across large memory horizons. Our experimental results verify this statement. Our ablation studies in Appx. D show that SIREN kernels consistently outperform all other variants. In addition, our experimental results in Sec. 5 show that *shallow* CKCNNs with SIREN kernels achieve state-of-the-art across datasets of different nature, i.e., with continuous and discrete data.

**The success of** Sine **nonlinearites: A spline basis interpretation.** Sitzmann et al. (2020) motivates the usage of Sine nonlinearities for implicit neural representations. However, there is no clear understanding as of *why* Sine nonlinearities are better suited for this task than (smooth) piece-wise nonlinearities. For the interested reader, we provide an interpretation to this phenomenon from a spline function approximation perspective in Appx. B.

Of most practical relevance from this analysis is the observation that proper initialization of the network parameters, particularly of the bias term $\{\mathbf{b}^{(l)}\}$, is important to create a well-spread set of basis functions suited for function approximation. For SIRENs, this is achieved by initializing the bias term uniformly across the period of each of the Sine components: $\mathbf{b}_i \sim \mathcal{U}(-\pi\|\mathbf{W}_{i,:}\|^{-1}, \pi\|\mathbf{W}_{i,:}\|^{-1})$. We observe that this initialization leads to better results and faster convergence for all tasks considered.

## 5 EXPERIMENTS

We validate our approach against several existing models and across several tasks selected from the corresponding papers. Specifically, we benchmark its ability to handle long-term dependencies, data at different resolutions and irregularly-sampled data. A complete description of the datasets used as well as additional experiments and ablation studies can be found in the Appendix (Appx. C, D). [1]

**Network details.** We parameterize our convolutional kernels as 3-layer SIRENs. Weight normalization (Salimans & Kingma, 2016) leads to better and faster convergence when applied to the layers in MLP, and we use it across all experiments. All our CKCNNs follow the structure shown in Fig. 8 and vary only in the number of blocks and channels. We use two residual blocks for all experiments reported in this section. Specifications on the architectures and hyperparameters used are given in Appx. E. We speed up the convolution operations in our networks via the *convolution theorem*: $(f * \psi) = \mathcal{F}^{-1}\{\mathcal{F}\{f\} \cdot \overline{\mathcal{F}\{\psi\}}\}$, with $\mathcal{F}$ the Fourier transform.

**Stress experiments.** First, we validate that CKCNNs can readily model memory horizons of different lengths. To this end, we evaluate if a shallow CKCNN is able to solve the *Copy Memory* and *Adding Problem* tasks (Hochreiter & Schmidhuber, 1997) for sequences of sizes in the range $[100, 6000]$. Success is achieved if 100% accuracy, or a loss $\leq$ 1e-4 are obtained, for the copy memory and adding problem, respectively. Random predictions for the adding problem lead to a loss of approx. 0.17.

Our results show that a shallow CKCNN is able to solve both problems for all sequence lengths considered without requiring structural modifications (Tab. 2). Recurrent architectures are not able not solve the copy problem at all and could solve the sum problem up to 200 steps. TCNs with $k$=7, $n$=7 were able to solve both tasks for up to 1000 steps. However, larger lengths were out of reach as their memory horizon is constrained a priori. To handle larger sequences, TCNs must modify their network structure based on prior knowledge regarding the expected length of the input sequence.

**Discrete sequences.** The continuous nature of our kernels might give the impression that CKCNNs are only suited for continuous data, i.e., time-series. However, Sine nonlinearities allow our convolutional kernels to model complex non-smooth non-linear functions (Fig. 4). Consequently, we validate whether CKCNNs can be applied for discrete sequence modeling on the following tasks: *sMNIST*, *pMNIST* (Le et al., 2015), *sCIFAR10* (Trinh et al., 2018) and *Char-level PTB* (Marcinkiewicz, 1994).

Shallow CKCNNs outperform recurrent, self-attention and convolutional models on sMNIST and pMNIST (Tab. 1). On sMNIST, a small CKCNN (100K params.) achieves state-of-the-art results with a model 80× smaller than the current state-of-the-art. A wider CKCNN (1M params.) slightly increases this result further. On pMNIST, we see an improvement of 0.8% over the best model of size ≤100K, and our wider shallow CKCNN achieves state-of-the-art on this dataset. For sCIFAR10,

---

[1]Our code is publicly available at `https://github.com/dwromero/ckconv`.

Table 1: Test results on discrete sequential datasets.

| MODEL | SIZE | sMNIST Acc (%) | pMNIST Acc (%) | sCIFAR10 Acc (%) | CHAR-PTB bpc |
|---|---|---|---|---|---|
| TCN (Bai et al., 2018a) | 70K | **99.0** | **97.2** | - | 1.31[†] |
| LSTM (Bai et al., 2018a) | 70K | 87.2 | 85.7 | - | 1.36[†] |
| GRU (Bai et al., 2018a) | 70K | 96.2 | 87.3 | - | 1.37[†] |
| IndRNN (Li et al., 2018) | 83K | 99.0 | 96.0 | - | - |
| DilRNN (Chang et al., 2017) | 44K | 98.0 | 96.1 | - | - |
| HiPPO (Gu et al., 2020a) | 0.5M | - | **98.30** | - | - |
| r-LSTM (Trinh et al., 2018) | 0.5M | 98.4 | 95.2 | 72.2 | - |
| Self-Att. (Trinh et al., 2018) | 0.5M | 98.9 | 97.9 | 62.2 | - |
| TrellisNet (Bai et al., 2018b) | 8M | 99.20 | 98.13 | **73.42** | 1.158[†] |
| CKCNN | 98K | 99.31 | 98.00 | 62.25 | - |
| CKCNN-Big | 1M | **99.32** | **98.54** | 63.74 | **1.045**[†] |

[†] Model sizes are 3M for TCN, LSTM and GRU, 13.4M for TrellisNet and 1.8M for CKCNN-Big.

Table 2: Evaluation on stress tasks. ✓ marks if the problem has been solved.

| MODEL | SIZE | SEQ. LENGTH | | | | |
|---|---|---|---|---|---|---|
| | | 100 | 200 | 1000 | 3000 | 6000 |
| | | COPY MEMORY | | | | |
| GRU | 16K | - | - | - | - | - |
| TCN | 16K | ✓ | ✓ | ✓ | - | - |
| CKCNN | 16K | ✓ | ✓ | ✓ | ✓ | ✓ |
| | | ADDING PROBLEM (LOSS) | | | | |
| GRU | 70K | ✓ | ✓ | - | | |
| TCN | 70K | ✓ | ✓ | ✓ | - | - |
| CKCNN | 70K | ✓ | ✓ | ✓ | ✓ | ✓ |

Table 3: Test accuracies on CT, SC and SC_raw.

| MODEL | SIZE | CT | SC | SC_RAW |
|---|---|---|---|---|
| GRU-ODE (De Brouwer et al., 2019) | 89K | 96.2 | 44.8 | ~ 10.0 |
| GRU-Δt (Kidger et al., 2020) | 89K | 97.8 | 20.0 | ~ 10.0 |
| GRU-D Che et al. (2018) | 89K | 95.9 | 23.9 | ~ 10.0 |
| ODE-RNN (Rubanova et al., 2019) | 88K | 97.1 | 93.2 | ~ 10.0 |
| NCDE (Kidger et al., 2020) | 89K | 98.8 | 88.5 | ~ 10.0 |
| CKCNN | 100K | **99.53** | **95.27** | **71.66** |

our small CKCNN obtains similar results to a self-attention model 5× bigger, and our wider variant improves performance by an additional 1%. Our best results are obtained with an even wider model (2.5M params) with which an accuracy of 65.59% is obtained. On Char-level PTB a CKCNN with 3M parameters outperforms all models considered as well as the state-of-the-art: Mogrifier LSTMs (Melis et al., 2019), while being 13.3× smaller.

**Time-series modeling.** Next, we evaluate CKCNNs on time-series data. To this end, we consider the *CharacterTrajectories (CT)* (Bagnall et al., 2018) and the *Speech Commands (SC)* (Warden, 2018) datasets. We follow Kidger et al. (2020) to obtain a balanced classification dataset with precomputed mel-frequency cepstrum coefficients. In addition, we evaluate the ability of CKCNNs to model long-term dependencies by training on the raw SC dataset (*SC_raw*), whose records have length 16k.

We compare CKCNNs with representative sequential models with continuous-time interpretations: GRU-ODE (De Brouwer et al., 2019), GRU-Δt (Kidger et al., 2020), ODE-RNN (Rubanova et al., 2019), and NCDE (Kidger et al., 2020). Continuous-time sequential models were selected as they are only sequential methods also able to handle irregularly-sampled data, and data at different resolutions. Our results show that shallow CKCNNs outperform all continuous-time models considered for both the CT and SC datasets (Tab. 3). In addition, CKCNNs obtain promising results on SC_raw, which validates their ability to handle very-long-term dependencies. In fact, CKCNNs trained on SC_raw are able outperform several Neural ODE models trained on the preprocessed data (SC).

In addition, we observed that neural ODE methods considered in Tab. 3 were prohibitively slow for long sequences. For instance, NCDEs were 228× slower than a CKCNN of equivalent size on SC_raw, taking 17 hours per epoch to train. Consequently, training a NCDE on SC_raw for a matching number of epochs would take more than 212 days to conclude. In order to provide results for these models, we train them under the same computational budget than CKCNNs. This is enough to train them for a single epoch. All obtained results are at best only marginally better than random.

**Testing at different sampling rates.** We now consider the case where a network is trained with data at a sampling rate $sr_1$, and tested with data at a different sampling rate $sr_2$. Our results show that the performances of CKCNNs remains stable for large sampling rate fluctuations (Tab. 5). This behaviour contrasts with most previous continuous-time models, whose performance rapidly decays upon these changes. CKCNNs outperform HiPPO (Gu et al., 2020a) and set a new state-of-the-art in this setting. Importantly, depending on the sampling, additional care may be needed to account for spatial displacements and high-frequencies of our kernels (see Appx. E.2 for details).

Table 4: Test results on irregular data.

| MODEL | PHYSIONET | CHARACTERTRAJECTORIES | | | | SPEECHCOMMANDS_RAW | | | |
|---|---|---|---|---|---|---|---|---|---|
| | AUC | (0%) | (30%) | (50%) | (70%) | (0%) | (30%) | (50%) | (70%) |
| GRU-ODE | 0.852 | 96.2 | 92.6 | 86.7 | 89.9 | $\sim 10.0$ | $\sim 10.0$ | $\sim 10.0$ | $\sim 10.0$ |
| GRU-$\Delta t$ | 0.878 | 97.8 | 93.6 | 91.3 | 90.4 | | | | |
| GRU-D | 0.871 | 95.9 | 94.2 | 90.2 | 91.9 | | | | |
| ODE-RNN | 0.874 | 97.1 | 95.4 | 96.0 | 95.3 | $\vdots$ | $\vdots$ | $\vdots$ | $\vdots$ |
| NCDE | 0.880 | 98.8 | 98.7 | **98.8** | **98.6** | | | | |
| CKCNN | **0.895** | **99.53** | **99.30** | **98.83** | 98.14 | 71.66 | 63.46 | 60.55 | 57.50 |

Table 5: Results for different train and test resolutions. Fractions depict resolutions proportional to the original one of the dataset. The accuracy of all models on the original resolution surpasses 90%.

| | **CKCNN – SIZE=100K** | | | | | |
|---|---|---|---|---|---|---|
| DATASET | TRAIN FREQ. | TEST FREQ. | | | | |
| | | 1 | $1/2$ | $1/4$ | $1/8$ | $1/16$ |
| CT | 1 | **99.53** | 99.30 | 99.30 | 95.80 | 76.45 |
| | $1/2$ | 98.83 | **99.07** | 98.37 | 96.97 | 80.42 |
| | $1/4$ | 96.74 | 96.97 | **99.30** | 98.83 | 84.85 |
| | $1/8$ | 96.97 | 97.44 | 97.20 | **99.30** | 73.43 |
| SC_RAW | 1 | **71.66** | 65.96 | 52.11 | 40.33 | 30.87 |
| | $1/2$ | 72.09 | **72.06** | 69.03 | 63.00 | 29.67 |
| | $1/4$ | 68.25 | 68.40 | **69.47** | 67.09 | 37.91 |
| | $1/8$ | 40.48 | 42.00 | 54.91 | **66.44** | 22.29 |

| **MODEL COMPARISON - CHARACTER TRAJECTORIES** | | | | | | |
|---|---|---|---|---|---|---|
| MODEL | GRU-D | ODE-RNN | LMU | NCDE | HIPPO | CKCNN |
| $1 \rightarrow 1/2$ | 23.1 | 41.8 | 44.7 | 6.0 | 88.8 | **99.30** |
| $1/2 \rightarrow 1$ | 25.5 | 31.5 | 11.3 | 13.1 | 90.1 | **98.83** |

**Irregularly-sampled data.** To conclude, we validate CKCNNs for irregularly-sampled data. To this end, consider the PhysioNet sepsis challenge (Reyna et al., 2019) as well as the CT dataset with drops of 30%, 50% and 70% of the data as in Kidger et al. (2020). In addition, we provide results under the same methodology for the SC_raw dataset. As in Kidger et al. (2020) we add an additional channel to the input to indicate whether the value at that position is known.

Our results show that CKCNNs outperform NCDEs and obtain state-of-the-art performance on the PhysioNet dataset. In addition, CKCNNs exhibit stable performance for varying quantities of missing data, and perform better than several models explicitly developed to this end (Tab. 4). On the CT dataset, NCDEs perform slightly better than CKCNNs for large data drop rates. However, we argue that our method is still advantageous due to the gains in training speed –see Section 6 for details–.

## 6 DISCUSSION AND LIMITATIONS

**Parameter-efficient large convolutional kernels.** CKConvs construct large complex kernels with a fixed parameter budget. For large input sequences, this results in large savings in the number of parameters required to construct global kernels with conventional CNNs. For sequences from the pMNIST (length = 784) and SC_raw (length = 16000) datasets, a conventional CNN with global kernels would require 2.14M and 46.68M of parameters, respectively, for a model equivalent to our CKCNN (100K). In other words, our kernel parameterization allows us to construct CKCNNs that are $21, 84$ and $445, 71$ times smaller than corresponding conventional CNNs for these datasets. Detailed exploration on the effect of our efficient continuous kernel parameterizations in optimization, overfitting and generalization is an interesting direction for future research.

**Is depth important? Shallow global memory horizons.** Our results are obtained with CKCNNs built with two residual blocks only. Additional experiments (Appx. D.2) indicate that our models do not benefit from larger depth, and suggest that CKCNNs do not rely on very deep features. Though further analysis is required to draw consistent conclusions, it is intriguing to explore if it is sufficient to equip neural networks with global memory horizons even if this happens in a shallow manner.

**High-frequency components.** Interestingly, our kernels often contain frequency components higher than the resolution of the grid used during training (Fig. 9). As a result, transitions to finer resolutions benefit from smoothing (see Appx. E.3). Nevertheless, we believe that, if tuned properly, these high-frequency components might prove advantageous for tasks such as super-resolution and compression.

**Faster continuous-time models.** CKCNNs rely on convolutions, and thus can be executed in parallel. As a result, CKCNNs can be trained faster than recurrent architectures. This difference becomes more pronounced with concurrent continuous-time models for sequential data, which are based on neural ODEs and require at least 5× slower than RNNs (Kidger et al., 2020). At the cost of larger memory costs, CKCNNs can be further sped up by using the convolution theorem.

**Neural networks parameterizing spatial functions should be able to model high-frequencies.** Our findings indicate that, common nonlinearities do not provide MLPs modelling spatial continuous functions the ability to model high-frequencies. Consequently, architectures that model continuous spatial functions via neural networks should transition towards models endowed with this ability, e.g., MLPs with Sine nonlinearities. These models encompass convolutional networks with continuous kernels, e.g., Schütt et al. (2017); Thomas et al. (2018); Wu et al. (2019), positional encodings in transformers, e.g., Romero & Cordonnier (2020); Hutchinson et al. (2020), and graph neural networks, e.g., Defferrard et al. (2020). Sine nonlinearities can be used to reduce the number of parameters needed to model local functions, or to extend the receptive field of the operations efficiently.

**Memory requirements.** Although, CKCNNs can be deployed and trained in parallel, CKCNNs must store the convolution responses at each layer and for all input positions. This induces a linear memory complexity with regard to the sequence length, and largely contrasts with recurrent continuous-time models, whose memory complexity is constant. The memory consumption of the operation is further incremented if the convolution theorem is applied because it requires multiplying the Fourier transform of the convolution and the kernel, and taking them back to the temporal representation. On the other hand, large convolutional kernels seem to allow CNNs to perform well without using many layers, which has a positive effect on memory consumption.

**Selection of $\omega_0$.** We observe that CKCNNs are very susceptible to the selection of $\omega_0$. For instance, performance on pMNIST may vary from $98.54$ to $65.22$ for values of $\omega_0$ in $[1, 100]$. Consequently, finding a good value of $\omega_0$ induces an important cost in hyperpararameter search (see Appx. E.4). $\omega_0$ acts as a prior on the variability of the target function. However, it is not obvious which value of $\omega_0$ is optimal for the internal (unknown) features of a network. Learning layer-wise $\omega_0$ values yielded sub-optimal results, and better results were obtained by using a predefined $\omega_0$ value across all layers.

## 7 CONCLUSION AND FUTURE WORK

We introduced the Continuous Kernel Convolution (CKConv), a simple, yet powerful approach able to model global long-term dependencies effectively in a parameter-efficient manner. Aside from the ability to get good accuracy, CKConvs are readily able to handle irregularly-sampled data, and data at different resolutions. CKCNNs achieve state-of-the-art results on multiple datasets, and often surpass neural architectures designed for particular settings, e.g., for irregularly-sampled data.

We are intrigued about the potential of CKCNNs for tasks in which (global) long-term dependencies play a crucial role, e.g., audio, video, reinforcement learning, (autoregressive) generative modeling. The usage of CKConvs to model long-term interactions in images is also very promising. In addition, CKConvs provide a convenient way to study the effect of the receptive field size of convolutional architectures, as no network modifications are required for different sizes. Our findings may also be useful for specific problems with irregularly-sampled data, e.g., medical, point clouds. We are also excited about structural advances of CKConvs. For instance, attentive versions of CKCNNs, or formulations that further improve computation and parameter efficiency

**Alleviating limitations.** Reducing the memory consumption of CKConvs is vital for its application on a broad range of scenarios, e.g., embedded devices. Moreover, finding kernel parameterizations more stable to hyperparameter changes is desirable to reduce the need for hyperparameter search.

**What is the best implicit kernel parameterization for convolutional kernels?** Despite the success of SIRENs, we believe that better kernel parameterizations might still be constructed, e.g., with Random Fourier Features (Tancik et al., 2020). Aside from improvements in implicit neural representations, which are directly transferable to CKConvs, we consider important to analyze the effect that having unknown, changing target objectives has on the approximation. A thorough empirical study of possible kernel parameterizations is an important direction for future research. A parameterization with which additional desiderata, e.g., smoothness, can be imposed is also desirable.

## REPRODUCIBILITY STATEMENT

We believe in reproducibility. In order to make our paper reproducible, we have release the source code used in our experiments to the public. In addition to the code, our repository includes the explicit command lines used to execute each of our experiments, as well as the corresponding pretrained models. Appx. E provides the experimental details of our approach. This section includes details regarding the hardware used, the specification of neural architecture as well as the inputs of $\texttt{MLP}^{\psi}$. It also states the method used for hyperparameter tuning and the hyperparameters of our final models. Details regarding the smoothing of high-frequency artifacts are also provided in this section. Details regarding the datasets and any preprocessing steps used are provided in Appx. C. The proofs of our claims can be found in Appx. A.

## ACKNOWLEDGEMENTS

We gratefully acknowledge Gabriel Dernbach for interesting analyses on the knot distribution of ReLU networks. We thank Emiel van Krieken and Ali el Hasouni as well for interesting questions and motivating comments at the beginning of this project.

David W. Romero is financed as part of the Efficient Deep Learning (EDL) programme (grant number P16-25), partly funded by the Dutch Research Council (NWO) and Semiotic Labs. Anna Kuzina is funded by the Hybrid Intelligence Center, a 10-year programme funded by the Dutch Ministry of Education, Culture and Science through the Netherlands Organisation for Scientific Research. Erik J. Bekkers is financed by the research programme VENI (grant number 17290) funded by the Dutch Research Council. All authors sincerely thank everyone involved in funding this work.

This work was carried out on the Dutch national einfrastructure with the support of SURF Cooperative

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

# APPENDIX

## A PROPERTIES OF CKCONVS

### A.1 VERY IRREGULARLY SAMPLED DATA

CKConvs can readily handle irregularly-sampled, and partially observed data. This is a result of the convolutional kernel $\texttt{MLP}^\psi$ being able to be sampled at arbitrary positions. For very non-uniformed sampled inputs, however, the corresponding sampling of the convolutional kernel can provide a biased estimation of the operation. To overcome this, one can follow the strategy proposed by Wu et al. (2019), which we summarize here for completeness.

For very non-uniformly sampled inputs, the continuous convolution $(x * \psi)(t) = \int_\mathbb{R} x(\tau)\psi(t-\tau)\,\mathrm{d}\tau$, must account for the distribution of samples in the input. Specifically, it is reformulated as:

$$(x * \psi)(t) = \int_\mathbb{R} s(\tau)x(\tau)\psi(t-\tau)\,\mathrm{d}\tau, \tag{8}$$

where $s(\tau)$ depicts the inverse sample density of the input at point $\tau$. Intuitively, $s(\tau)$ controls the contribution of points $x(\tau)$ to the output response. If multiple points are close to one another, their contribution should be smaller than the contribution of points in regions where the sample distribution is much sparser. This provides a Monte Carlo estimate of $(x * \psi)$ from biased samples. In particular, one has that:

$$\int f(\tau)\,\mathrm{d}\tau = \int \frac{f(\tau)}{p(\tau)}p(\tau)\,\mathrm{d}\tau \approx \sum_i \frac{f(\tau_i)}{p(\tau_i)}, \text{ for } \tau_i \sim p(\tau).$$

With $s(\tau) = \frac{1}{p(\tau)}$, Eq. 8 provides an unbiased estimation of $(x * \psi)$.

### A.2 DATA SAMPLED AT DIFFERENT SAMPLING RATES

In addition, CKConvs are readily able to handle data at different resolutions. In particular, the continuous kernel convolution between an input signal $\mathbf{x}$ and a continuous convolutional kernel $\boldsymbol{\psi}$ calculated at sampling rates $\mathrm{sr}_1$: $(\mathbf{x} * \boldsymbol{\psi})_{\mathrm{sr}_1}$, and $\mathrm{sr}_2$: $(\mathbf{x} * \boldsymbol{\psi})_{\mathrm{sr}_2}$, are approximately equal up to a normalization factor given by $\frac{\mathrm{sr}_2}{\mathrm{sr}_1}$:

$$(\mathbf{x} * \boldsymbol{\psi})_{\mathrm{sr}_2}(t) \approx \frac{\mathrm{sr}_2}{\mathrm{sr}_1}(\mathbf{x} * \boldsymbol{\psi})_{\mathrm{sr}_1}(t).$$

Consequently, CKCNNs *(i)* can be deployed at sampling rates different than those seen during training, and *(ii)* can be trained on data with varying spatial resolutions. The later is important for tasks in which data can be given at different resolutions such as super-resolution and segmentation.

*Proof.* To prove the previous statement, we start with the continuous definition of the convolution:

$$(x * \psi)(t) = \int_\mathbb{R} x(\tau)\psi(t-\tau)\,\mathrm{d}\tau,$$

where we assume for simplicity and without loss of generality that the functions $x$, $\psi$ are scalar-valued.

In practice, an integral on a continuous function $f : \mathbb{R} \to \mathbb{R}$ cannot be computed on finite time. Consequently, it is approximated via a Riemann integral defined on a finite grid $\{\tau_{\mathrm{sr},i}\}_{i=1}^{N_{\mathrm{sr}}}$ obtained by sampling $\tau$ at a sampling rate $\mathrm{sr}$:

$$\int f(\tau)\,\mathrm{d}\tau \approx \sum_{i=1}^{N_{\mathrm{sr}}} f(\tau_{\mathrm{sr},i})\Delta_{\mathrm{sr}},$$

where $\Delta_{\mathrm{sr}} = \frac{1}{\mathrm{sr}}$ depicts the distance between sampled points. For two sampling rates $\mathrm{sr}_1$, $\mathrm{sr}_2$, the convolution can be approximated through the corresponding Riemann integrals:

$$\int_\mathbb{R} x(\tau)\psi(t-\tau)\,\mathrm{d}\tau \approx \sum_{i=1}^{N_{\mathrm{sr}_1}} x(\tau_{\mathrm{sr}_1,i})\psi(t-\tau_{\mathrm{sr}_1,i})\Delta_{\mathrm{sr}_1}$$

$$\approx \sum_{i=1}^{N_{\mathrm{sr}_2}} x(\tau_{\mathrm{sr}_2,i})\psi(t-\tau_{\mathrm{sr}_2,i})\Delta_{\mathrm{sr}_2}$$

As a result, we have that both approximations are approximately equal to the continuous integral at positions $t$ defined on both discrete grids. By equating both approximations, we obtain that:

$$\sum_{i=1}^{\mathrm{N_{sr_2}}} x(\tau_{\mathrm{sr_2},i})\psi\big(t - \tau_{\mathrm{sr_2},i}\big)\Delta_{\mathrm{sr_2}} \approx \sum_{i=1}^{\mathrm{N_{sr_1}}} x(\tau_{\mathrm{sr_1},i})\psi\big(t - \tau_{\mathrm{sr_1},i}\big)\Delta_{\mathrm{sr_1}}$$

$$\underbrace{\sum_{i=1}^{\mathrm{N_{sr_2}}} x(\tau_{\mathrm{sr_2},i})\psi\big(t - \tau_{\mathrm{sr_2},i}\big)\frac{1}{\mathrm{sr_2}}}_{(x*\psi)_{\mathrm{sr_2}}(t)} \approx \underbrace{\sum_{i=1}^{\mathrm{N_{sr_1}}} x(\tau_{\mathrm{sr_1},i})\psi\big(t - \tau_{\mathrm{sr_1},i}\big)\frac{1}{\mathrm{sr_1}}}_{(x*\psi)_{\mathrm{sr_1}}(t)}$$

$$(x * \psi)_{\mathrm{sr_2}}(t) \approx \frac{\mathrm{sr_2}}{\mathrm{sr_1}}(x * \psi)_{\mathrm{sr_1}}(t)$$

which concludes the proof.

### A.3 LINEAR RECURRENT UNITS ARE CKCONVS

Interesting insights can be obtained by drawing connections between convolutions and recurrent units. In particular, we can show that linear recurrent units are equal to a CKConv with a particular family of convolutional kernels: exponential functions. Besides providing a generalization to recurrent units, this equality provides a fresh and intuitive view to the analysis of vanishing and exploding gradients.

**Recurrent unit.** Given an input sequence $\mathcal{X} = \{\mathbf{x}(\tau)\}_{\tau=0}^{\mathrm{N_X}}$, a recurrent unit is constructed as:

$$\mathbf{h}(\tau) = \sigma\big(\mathbf{W}\mathbf{h}(\tau - 1) + \mathbf{U}\mathbf{x}(\tau)\big) \tag{9}$$
$$\tilde{\mathbf{y}}(\tau) = \mathrm{softmax}(\mathbf{V}\mathbf{h}(\tau)), \tag{10}$$

where $\mathbf{U}, \mathbf{W}, \mathbf{V}$ parameterize the *input-to-hidden*, *hidden-to-hidden* and *hidden-to-output* connections of the unit. $\mathbf{h}(\tau), \tilde{\mathbf{y}}(\tau)$ depict the hidden representation and the output at time-step $\tau$, and $\sigma$ represents a point-wise non-linearity.

The hidden representation $\mathbf{h}$ of a *linear recurrent unit*, i.e., with $\sigma=\mathrm{Id}$, can be written as a convolution. To see this, consider the hidden representation of the unit unrolled for $t$ steps:

$$\mathbf{h}(t) = \mathbf{W}^{t+1}\mathbf{h}(-1) + \sum_{\tau=0}^{t} \mathbf{W}^{\tau}\mathbf{U}\mathbf{x}(t - \tau). \tag{11}$$

Here, $\mathbf{h}(-1)$ is the initial state of the hidden representation. We see that in fact it corresponds to a convolution between an input signal $\mathbf{x}$ and a convolutional kernel $\psi$ given by:[2]

$$\mathbf{x} = \big[\mathbf{x}(0), \mathbf{x}(1), ..., \mathbf{x}(t - 1), \mathbf{x}(t)\big] \tag{12}$$
$$\psi = \big[\mathbf{U}, \mathbf{W}\mathbf{U}, ..., \mathbf{W}^{t-1}\mathbf{U}, \mathbf{W}^{t}\mathbf{U}\big] \tag{13}$$
$$\mathbf{h}(t) = \sum_{\tau=0}^{t} \mathbf{x}(\tau)\psi(t - \tau) = \sum_{\tau=0}^{t} \mathbf{x}(t - \tau)\psi(\tau). \tag{14}$$

Drawing this equality yields some important insights:

**The cause of the exploding and vanishing gradients.** Eqs. 12-14 intuitively depict the root of the exploding and vanishing gradient problem. It stems from sequence elements $\mathbf{x}(t - \tau)$ $\tau$ steps back in the past being multiplied with an effective convolutional weight $\psi(\tau)=\mathbf{W}^{\tau}\mathbf{U}$. For eigenvalues of $\mathbf{W}$, $\lambda$, other than one, the resulting convolutional kernel $\psi$ can only represent functions that either grow ($\lambda\geq1$) or decrease ($\lambda\leq1$) exponentially as a function of the sequence length (Figs. 3a, 3b). As a result, the contribution of input values in the past either rapidly fades away or governs the updates of the model parameters. As exponentially growing gradients lead to divergence, the eigenvalues of $\mathbf{W}$ for converging architectures are often smaller than 1. This explains the effective small effective memory horizon of recurrent networks.

**Linear recurrent units are a subclass of CKConvs.** Linear recurrent units can be described as a convolution between the input and a very specific class of convolutional kernels: exponential functions (Eq. 13). In general, however, convolutional kernels are not restricted to this functional class. This can be seen in conventional (discrete) convolutions, whose kernels are able to model complex functions within their memory horizon. Unfortunately, discrete convolutions use a predefined, small kernel size,

---

[2] We discard $\mathbf{h}(-1)$ as it only describes the initialization of $\mathbf{h}$.

and thus possess a restricted memory horizon. This is equivalent to imposing an effective magnitude of zero to all input values outside the memory horizon (Fig. 3c). CKConvs, on the other hand, are able to define arbitrary large memory horizons. For memory horizons of size equal to the input length, CKConvs are able to model complex functions upon the entire input (Fig. 3d).

In conclusion, we illustrate that CKConvs are also a generalization of (linear) recurrent architectures which allows for parallel training and enhanced expressivity.

## B  AN SPLINE INTERPRETATION OF ReLU AND SINE NETWORKS

Sitzmann et al. (2020) motivates the usage of Sine nonlinearities for implicit neural representations. However, there is no clear understanding as of *why* Sine nonlinearities are better suited for this task than (smooth) piece-wise nonlinearities. Here, we provide an interpretation to this phenomenon from a spline function approximation perspective.

### B.1  KERNEL PARAMETERIZATION VIA ReLU NETWORKS

**The importance of initialization.**  There is an important distinction between implicit neural representations and conventional neural applications regarding the assumed distribution of the input. Conventional applications assume the distribution of the input features to be centered around the origin. This is orthogonal to implicit neural representations, where the spatial distribution of the output, i.e., the value of the function being implicitly represented, is *uniformly distributed*.

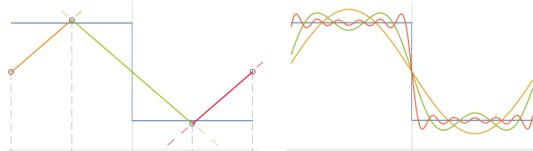

For ReLU networks, function approximation is equivalent to an approximation via a max-spline basis (Balestriero & Baraniuk, 2018), and its expressiveness is determined by the number of *knots* the basis provides, i.e., places where a non-linearity bends the space. Naturally, the better the placing of these knots at initialization, the faster the approximation may converge. For applications in which the data is centered around zero, initializing the knots around zero is a good inductive bias.[3] However, for spatially uniform distributed inputs, the knots should be uniformly distributed (Fig. 5). As a result, conventional initializations lead to very poor reconstructions (ReLU 0-Init, Fig. 4), and explicitly aggregating positional encodings to the mappings leads to important improvements, e.g, Mildenhall et al. (2020).

Figure 5: An step function approximated via a spline basis (left) and a periodic basis (right). As the target function is defined uniformly on a given interval, uniformly initializing the knots of the spline basis provides faster and better approximations. Periodic bases, on the other hand, periodically bend space, and thus can be tuned easier to approximate the target function at arbitrary points in space.

For ReLU layers $\mathbf{y}=\max\{\mathbf{0}, \mathbf{W}\mathbf{x} + \mathbf{b}\}$ knots appear at the point where $\mathbf{0}=\mathbf{W}\mathbf{x} + \mathbf{b}$. To place the knots at $\mathbf{x}=\mathbf{0}$, it is sufficient to set the bias to zero: $\mathbf{b}=\mathbf{0}$. For uniformly distributed knots in a range $[\mathbf{x}_{\min}, \mathbf{x}_{\max}]$, however, one must solve the ReLU equation for uniformly distributed points in that range: $\mathbf{0}=\mathbf{W}\mathbf{x}_{\text{unif}} + \mathbf{b}$. It results that $\mathbf{b}=-\mathbf{W}\mathbf{x}_{\text{unif}}$, for arbitrary values of $\mathbf{W}$.

In multilayered networks, the approximation problem can be understood as reconstructing the target function in terms of a basis $\mathbf{h}^{(L-1)}$. Consequently, the expressivity of the network is determined by the number of knots in $\mathbf{h}^{(L-1)}$. In theory, each ReLU layer is able to divide the linear regions of the previous layer in exponentially many sub-regions (Montufar et al., 2014; Serra et al., 2018), or equivalently, to induce an exponential layer-wise increase in the number of knots. For the first layer, the positions of the knots are described by the bias term, and for subsequent layers, these positions also depend on $\mathbf{W}^{(l)}$. Unfortunately, as depicted by Hanin & Rolnick (2019), slight modifications of $\{\mathbf{W}^{(l)}, \mathbf{b}^{(l)}\}$ can strongly simplify the landscape of the linear regions, and thus the knots (Fig. 6). More importantly, Hanin & Rolnick (2019) showed that the number of linear regions at initialization is actually equal to a constant times the number of neurons in the network (with a constant very close

---

[3]This is why $\mathbf{b}=\mathbf{0}$ is common in regular initialization schemes.

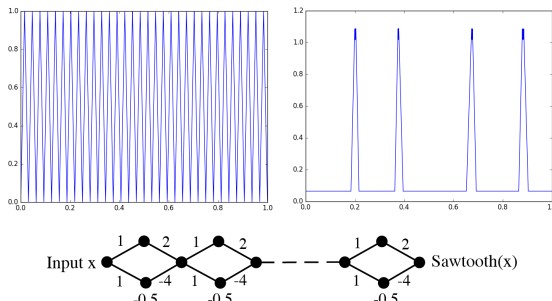

Figure 6: The sensitivity of networks with layer-wise exponential growing to slight changes. Taken from Hanin & Rolnick (2019). The sawtooth function with $2^n$ teeth (left) can be easily expressed via a ReLU network with $3n + 4$ neurons (bottom). However, a slight perturbation of the network parameters –Gaussian noise with standard deviation $0.1$– greatly simplifies the linear regions captured by the network, and thus the distribution of the knots in the basis (right).

to one in their experiments). In addition, they show that this behavior barely changes throughout training.

**An improved initialization scheme.** Following the previous reasoning, we explore inducing a uniformly distributed initialization of the knots. However, we observe that finding an initialization with an exponential number of knots is a cumbersome and unstable procedure. In fact, it is not always possible, and, whenever possible, it strongly restricts the values the weights $\mathbf{W}^{(l)}$ can assume.

Following the findings of Hanin & Rolnick (2019), we instead employ an initialization procedure with which the total number of knots is equal to the number of neurons of the network. This is obtained by replicating the initialization procedure of the first layer throughout the network: For randomly initialized weights $\mathbf{W}^{(l)}$, the bias term $\mathbf{b}^{(l)}$ is given by the equality $\mathbf{b}^{(l)} = -\mathbf{W}^{(l)}\mathbf{h}^{(l)}(\mathbf{x}_{\text{unif}})$, where $\mathbf{x}_{\text{unif}}$ is a vector of uniformly distributed points in $[\mathbf{x}_{\min}, \mathbf{x}_{\max}]$. Interestingly, we observe that this initialization strategy consistently outperforms the standard initialization for a large range of target functions (ReLU Unif-Init, Fig. 7). Unfortunately however, we note that ReLU networks still show large difficulties in representing very nonlinear and non-smooth functions. In Fig. 7, we illustrate that other popular nonlinearities: LeakyReLU, Swish, exhibit the same behavior.

### B.2 KERNEL PARAMETERIZATION VIA Sine NETWORKS

Recently, Sitzmann et al. (2020) proposed to replace ReLU nonlinearities by Sine for the task of implicit neural representation learning. Based on their relation with implicit neural representations, we explore using Sine networks to parameterize our continuous kernels. Intriguingly, we observe that this slight modification allows our kernels to approximate *any* provided function to near perfection, and leads to a consistent improvement for all tasks considered in this paper (Appx. D.1, Fig. 7).

A possible explanation to these astonishing results can be provided via our prior analysis:

**Periodic bending of the space.** A Sine layer is given by: $\mathbf{y} = \text{Sin}(\omega_0[\mathbf{W}\mathbf{x} + \mathbf{b}])$, where $\omega_0$ works as a prior on the variability of the target function. Orthogonal to ReLU layers, Sine layers periodically bend the space. As a result, the same $\mathbf{y}$ value is obtained for all bias values $\mathbf{b}'_i = \mathbf{b}_i + n2\pi\|\mathbf{W}_{i,:}\|^{-1}$, $\forall n \in \mathbb{Z}$. This is important from a spline approximation perspective. While for ReLU layers a unique value of $\mathbf{b}$ exists that bends the space at a desired position, infinitely many values of $\mathbf{b}$ do so for Sine ones. Resultantly, Sine layers are much more robust to parameter selection, and can be tuned to benefit pattern approximation at arbitrary –or even multiple– positions in space (Fig. 5, right). We conjecture that this behavior leads to much more reliable approximations and faster convergence.

**An exponentially big Fourier basis.** It is not surprising for a (large) basis of phase-shifted sinusoidal functions to be able to approximate arbitrary functions with high fidelity. This result was first observed over two centuries ago by Fourier (1807) and lies at the core of the well-known *Fourier transform*: any integrable function can be described as a linear combination of a (possibly) infinite basis of phase-shifted sinusoidal functions. Sitzmann et al. (2020) proposed an initialization of $\{\mathbf{W}^{(l)}\}$ that allows for the construction of deep Sine networks able to periodically divide the space into

exponentially many regions as a function of depth. Intuitively, approximations via Sine networks can be seen in terms of an exponentially large Fourier-like basis. We conjecture that this exponential growth combined with the periodicity of sine is what allows for astonishingly good approximations: the more terms in a Fourier transform, the better the approximation becomes.

Interestingly, we find that a uniformly distributed initialization of the bias term $\mathbf{b}_i \sim \mathcal{U}(-\pi\|\mathbf{W}_{i,:}\|^{-1}, \pi\|\mathbf{W}_{i,:}\|^{-1})$ also leads to better and faster convergence for Sine networks.

## C  DATASET DESCRIPTION

**Copy Memory Problem.** The copy memory task consists of sequences of length $T{+}20$, for which the first 10 values are chosen randomly among the digits $\{1, ..., 8\}$, the subsequent $T{-}1$ digits are set to zero, and the last 11 entries are filled with the digit 9. The goal is to generate an output of the same size of the input filled with zeros everywhere except for the last 10 values, for which the model is expected to predict the first 10 elements of the input sequence.

**The Adding Problem.** The adding problem consists of input sequences of length $T$ and depth 2. The first dimension is filled with random values in $[0, 1]$, whereas the second dimension is set to zeros except for two elements marked by 1. The objective is to sum the random values for which the second dimension is equal to 1. Simply predicting the sum to be 1 results in a MSE of about 0.1767.

**Sequential and Permuted MNIST.** The MNIST dataset (LeCun et al., 1998) consists of 70K grayscale 28×28 handwritten digits divided into training and test sets of 60K and 10K samples, respectively. The sequential MNIST dataset (sMNIST) presents MNIST images as a sequence of 784 pixels for digit classification. Consequently, good predictions require preserving long-term dependencies up to 784 steps in the past: much longer than most language modelling tasks (Bai et al., 2018b).

The permuted MNIST dataset (pMNIST) additionally permutes the order or the sMNIST sequences at random. Consequently, models can no longer rely on on local features to perform classification. As a result, the classification problem becomes more difficult and the importance of long-term dependencies more pronounced.

**Sequential CIFAR10.** The CIFAR10 dataset (Krizhevsky et al., 2009) consists of 60K real-world $32 \times 32$ RGB images uniformly drawn from 10 classes divided into training and test sets of 50K and 10K samples, respectively. Analogously to the sMNIST dataset, the sequential CIFAR10 (sCIFAR10) dataset presents CIFAR10 images as a sequence of 1024 pixels for image classification. This dataset is more difficult than sMNIST, as *(i)* even larger memory horizons are required to solve the task, and *(ii)* more complex structures and intra-class variations are present in the images (Trinh et al., 2018).

**CharacterTrajectories.** The CharacterTrajectories dataset is part of the UEA time series classification archive (Bagnall et al., 2018). It consists of 2858 time series of different lengths and 3 channels representing the $x, y$ positions and the pen tip force while writing a Latin alphabet character in a single stroke The goal is to classify which of the different 20 characters was written using the time series data. The maximum length of the time-series is 182.

**Speech Commands.** The Speech Commands dataset (Warden, 2018) consists of 105809 one-second audio recordings of 35 spoken words sampled at 16kHz. Following Kidger et al. (2020), we extract 34975 recordings from ten spoken words to construct a balanced classification problem. We refer to this dataset as **SC_raw**. In addition, we utilize the preprocessing steps of Kidger et al. (2020) and extract mel-frequency cepstrum coefficients from the raw data. The resulting dataset, named **SC**, consists of time series of length 161 and 20 channels.

**PhysioNet.** The PhysioNet 2019 challenge on sepsis prediction (Goldberger et al., 2000; Reyna et al., 2019) is a irregularly sampled, partially observed dataset consisting of 40335 time series of variable length describing the stay of patients within an ICU. Time-series are made out of 5 static features, e.g., age, and 34 time-dependent features, e.g., respiration rate, creatinine blood concentration, and 10.3% of the values are observed. We follow Kidger et al. (2020) and consider the first 72 hours of a patient's stay to predict whether sepsis is developed over the course of their entire stay –which can extend for a month for some patients–.

**PennTreeBank.** The PennTreeBank (PTB) (Marcinkiewicz, 1994) is a language corpus which consists of 5,095K characters for training, 396K for validation and 446K for testing. On a char

Table 6: Test accuracies of CKCNNs with multiple $\text{MLP}^\psi$ nonlinearities. Model size = 100K.

| NON-LINEARITY | DATASET | | | |
|---|---|---|---|---|
| | sMNIST | pMNIST | SC | SC_raw |
| ReLU | 81.21 | 59.15 | 94.97 | 49.15 |
| LeakyReLU | 80.57 | 55.85 | 95.03 | 38.67 |
| Swish | 85.20 | 61.77 | 93.43 | 62.23 |
| Sine | **99.31** | **98.00** | **95.27** | **71.66** |

Table 7: Test accuracy of CKCNNs for various depths and widths.

| | pMNIST | | | |
|---|---|---|---|---|
| DEPTH | FIXED WIDTH | | FIXED SIZE | |
| | Size | Acc.(%) | Size | Acc.(%) |
| 2 Blocks | 98k | 99.21 | 98k | **99.21** |
| 4 Blocks | 225k | **99.26** | 95k | **99.19** |
| 8 Blocks | 480k | **99.29** | 105k | 99.12 |
| 16 Blocks | 990k | 99.19 | 107k | 99.02 |

lever that we use in our experiment the vocabulary size is 50 characters (or the size of the alphabet, including end-of-string char). We follow Bai et al. (2018a) in performing character-level language modeling task on this dataset.

# D    ABLATION STUDIES

In this section, we perform an ablative study of our approach. Specifically, we analyze the effect of multiple components of our network, and provide additional comparisons with alternative architectures. Specifications on the architectures and hyperparameters used are given in Appx. E.

## D.1    USING SINE NON-LINEARITIES OVER POPULAR ALTERNATIVES

As shown in Sec. 4.2, Sine nonlinearities provide astonishing improvements over equivalent networks with ReLU nonlinearities for function reconstruction. In this section, we provide additional experiments to highlight the suitability of Sine nonlinearities over other popular alternatives both for function approximation and the rest of the tasks considered in this work. The same architectures are used across all experiments and vary only in the nonlinearity used in $\text{MLP}^\psi$. We find that nonlinearities other than Sine benefit from layer normalization and thus we incorporate it in these variants.

**Case I: Function Approximation via $\text{MLP}^\psi$.** First, we evaluate the problem of function approximation in Sec. 4.2, Fig. 4, for nonlinearities other than ReLU and Sine. In particular, we approximate several functions with a $\text{MLP}^\psi$ network which varies only in the type of nonlinearity used: ReLU (Nair & Hinton, 2010), LeakyReLU (Xu et al., 2015), Swish (Ramachandran et al., 2017), and Sine (Sitzmann et al., 2020).

Our results (Fig. 7), illustrate that Sine provides astonishing approximation capabilities over all other nonlinearities considered. In particular, we observe that Sine is the only nonlinearity able to reconstruct very nonlinear and very non-smooth functions, while all other alternatives fail poorly.

**Case II: CKCNNs with nonlinearities other than** Sine. Next, we consider the case in which CKCNNs with nonlinearities other than Sine are used to solve the tasks considered in Sec. 5. In particular, we train CKCNNs on sMNIST, pMNIST, SC and SC_raw for four different nonlinearities: ReLU, LeakyReLU, Swish, Sine. We utilize the same backbone architecture used in the main text for the corresponding dataset.

Our results (Tab. 6) indicate that Sine outperform CKCNNs using any of the other nonlinearities.

**Analysis of the results.** Our findings indicate Sine is much better suited to describe continuous spatial functions via neural networks than all other nonlinearities considered. This result motivates replacing popular nonlinearities by Sine for applications in which neural networks are used to describe continuous positional functions. This family of models encompass –but is not restricted to– continuous types of convolutions, e.g., Schütt et al. (2017); Thomas et al. (2018); Finzi et al. (2020); Fuchs et al. (2020), as well as positional encodings in transformers, e.g., Dai et al. (2019); Ramachandran et al. (2019); Romero & Cordonnier (2020), and graph neural networks, e.g., Defferrard et al. (2020). We consider this result to be of large relevance to the deep learning community.

## D.2    GOING DEEPER WITH CKCNNS

The experimental results shown in Sec. 5 are obtained with shallow CKCNNs composed of 2 residual blocks only. An interesting question is whether going deeper can be used to improve the performance

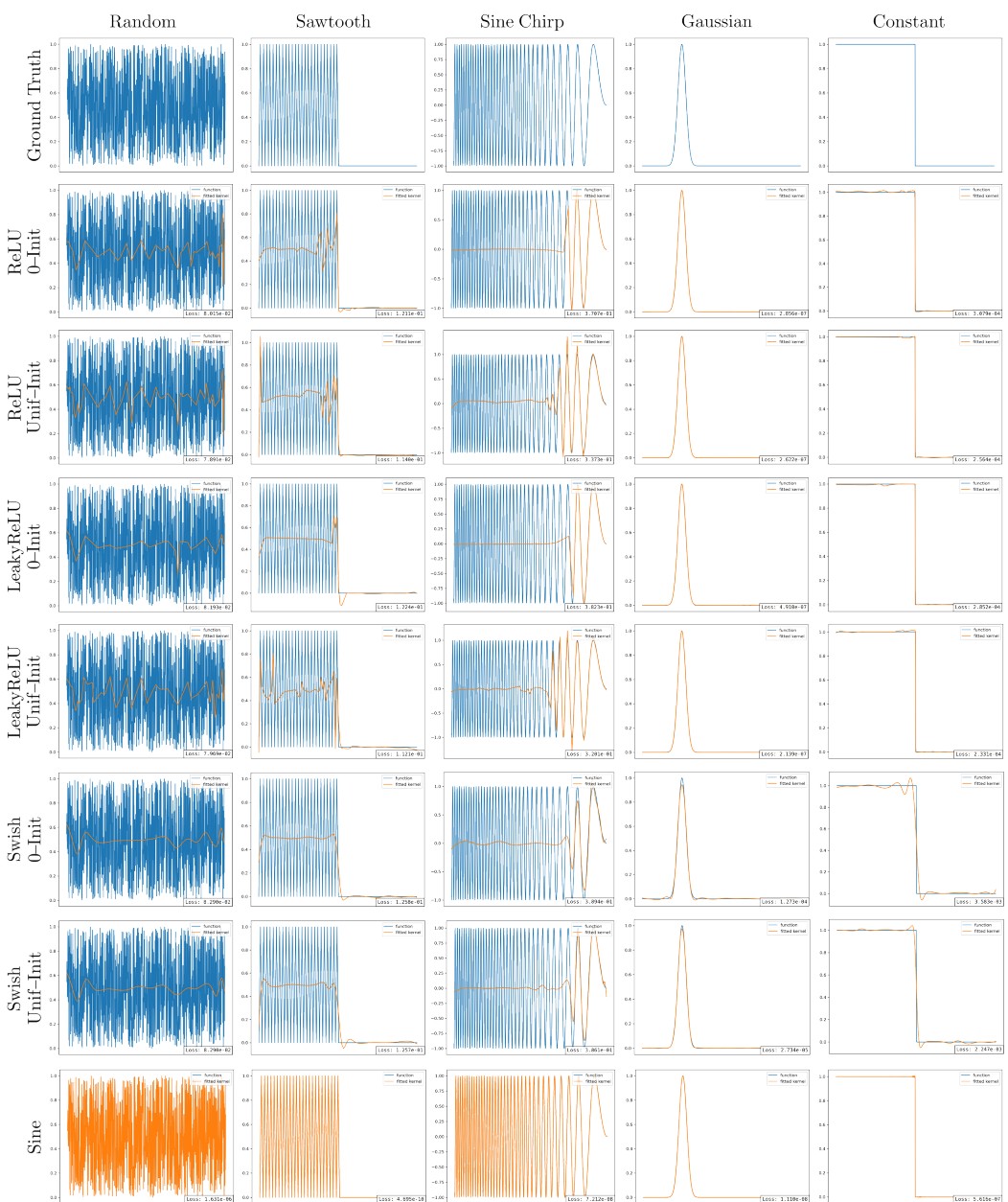

Figure 7: Function approximation via ReLU, LeakyReLU, Swish and Sine networks. All network variants perform a decent job in approximating simple functions. However, for non-linear, non-smooth functions, all networks using nonlinearities other than Sine provide very poor approximations. Interestingly, the uniform knot initialization proposed in Sec. 4.2 provides consistent improvements for all network variants. However, despite this improvement, the approximation results remain insufficient. Contrarily, Sine networks quickly and seamlessly approximate all functions. All network configurations are equal up to the non-linearities used.

of CKCNNs. To analyze this, we compare deep and shallow CKCNNs with the same architecture for equal width, and equal number of parameters.

Our results (Tab. 7) indicate that deep CKCNNs do not provide improvements over shallow CKCNNs. In fact, deep CKCNNs of fixed size underperform their shallow counterparts. This is an interesting results as shallow CKCNNs do not strongly rely on deep-wise compositionality of features, which is largely considered indispensable in deep learning.

**Analysis of the results.** The dynamics governing these results are not yet fully understood. However, our findings may lead to two different conclusions, both of which we consider important for the development and understanding of CKCNNs and deep learning in general:

*Outcome I: Deep CKCNNs.* The first possible outcome is that our current parameterization does not correctly leverage depth. In this case, efforts to construct proper *deep* CKCNNs will likely lead to performance improvements over the current architectures, and thus have the potential to advance the state-of-the-art further.

*Outcome II: Depth is not needed when global memory horizons are provided with shallow networks.* The second possible outcome is that depth is used mainly as a means to construct global memory horizons. Consequently, neural networks do not have to be very deep at all provided that global memory horizons are defined by shallow neural networks. Interestingly, this conclusion is in line with the predominant design of recurrent architectures, for which a moderate number of layers are used, e.g., Pascanu et al. (2013a); Graves et al. (2013); Gu et al. (2020b;a). This possible outcome is very exciting as depth is largely considered indispensable in the deep learning community.

# E EXPERIMENTAL DETAILS

In this section, we provide extended details over our implementation as well as the exact architectures and optimization schemes used in our experiments.

## E.1 GENERAL REMARKS

Our models follow the structure shown in Fig. 8 and vary only in the number of channels. We use layer normalization (Ba et al., 2016) in our backbone network, and use the Adam optimizer (Kingma & Ba, 2014) across all our experiments. Our code is implemented in `PyTorch` and is publicly available at *link removed for the sake of the double-blind review*. We utilize `wandb` (Biewald, 2020) to log our results, and use NVIDIA TITAN RTX GPUs throughout our experiments.

**Continuous Convolutional Kernel `MLP`$^\psi$.** All our convolutional kernels are parameterized by a vector-valued 3-layer neural network with 32 hidden units and $\mathrm{Sine}$ nonlinearities:

$$1 \to 32 \to 32 \to \mathrm{N_{out}} \times \mathrm{N_{in}},$$

where $\mathrm{N_{in}}$, $\mathrm{N_{Cout}}$ are the number of input and output channels of the convolutional layer. We utilize weight normalization (Salimans & Kingma, 2016) in our `MLP`$^\psi$ networks, and select a hidden size of 32 based on empirical evidence and findings from previous works, e.g., Finzi et al. (2020).

**Normalized relative positions.** The `MLP`s parameterizing our convolutional kenels receive relative positions as input. However, considering unitary step-wise relative positions, i.e., 0, 1, 2, ... , N, can be problematic from a numerical stability perspective as N may grow very large, e.g., N=16000 for the SC_raw dataset. Consequently, we follow good practices from works modelling continuous functions with neural networks, and map the largest unitary step-wise relative positions seen during training $[0, \mathrm{N}]$ to a uniform linear space in $[-1, 1]$.

**Hyperparameter tuning.** We tune the hyperparameters of our models via the `bayes` method given in `wandb` Sweeps, which selects hyperparameter values via a Gaussian process over the results obtained so far. We perform tuning on a validation dataset until a predefined maximum number of runs of 100 is exhausted. Further improvements upon our results may be obtained by leveraging more sophisticated tuning methods as well as additional runs.

**Selecting $\omega_0$.** CKCNNs are very susceptible to the value of $\omega_0$. In order to obtain a reasonable $\omega_0$, we first perform a random search on a large interval $\omega_0 \in [0, 3000]$. After a few runs, we stop the random search and select the subinterval in which the validation accuracy is most promising. Next,

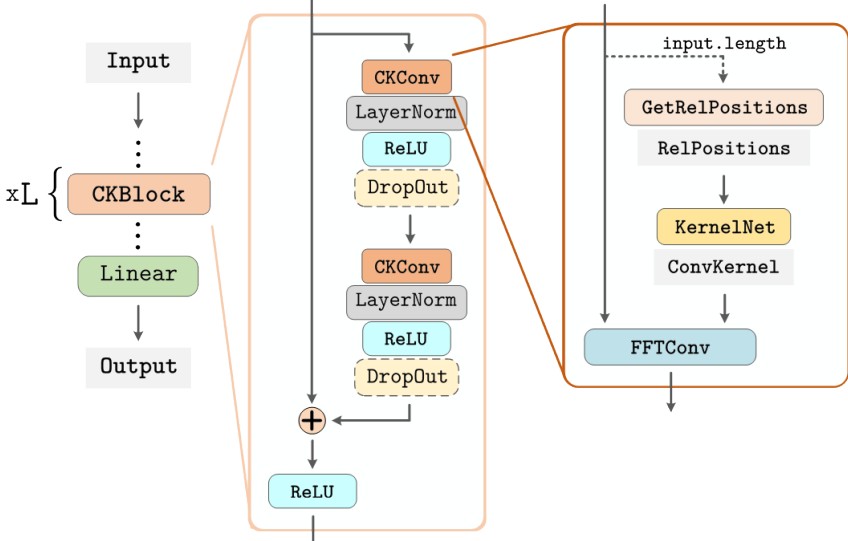

Figure 8: Graphical description of continuous kernel convolutional networks. Dot-lined blocks depict optional blocks, and blocks without borders depict variables. `KernelNet` blocks use Sine nonlinearities. We replace spatial convolutions by Fourier Convolutions (`FFTConv`), which leverages the convolution theorem to speed up computations.

we restart the random search on this sub-interval and repeat the process until a $\omega_0$ value is obtained, for which the validation accuracy is sufficiently high. Surprisingly, we found optimal values of $\omega_0$ to be always enclosed in the interval $[1, 70]$ even for very long sequences as in SC_raw.

### E.2 ACCOUNTING FOR SPATIAL DISPLACEMENTS OF THE SAMPLED CONVOLUTIONAL KERNELS

We follow the sampling procedure of Gu et al. (2020a) throughout our test sampling rate discrepancy experiments. Specifically, for a sequence `seq` of length N, subsampling by a factor `n` is performed by running `seq[::n]`. That is, by taking the `n`-th element of the sequence starting from its first element. For example, for a sequence of length N=182, different values of `n` would yield the following sequences:

```
(n = 1) → [1, 2, 3,   ...   , 180, 181, 182]
(n = 2) → [1, 3, 5,   ...   , 177, 179, 181]
(n = 4) → [1, 5, 9,   ...   , 173, 177, 181]
(n = 8) → [1, 9, 17, ...   , 161, 169, 177]
```

Recall that `MLP`$^\psi$ takes normalized relative positions in $[-1, 1]$ as input, which are computed based on the max input length seen during training. However, some of these subsampling transitions *change* the max value of the sequence, e.g., for (`n = 8`) the maximum is given by 177, whereas for (`n = 1`) this value corresponds to 182. Consequently, a naive approach would consider the last position in each subsampled sequence to correspond to the maximum normalized relative position 1. This effectively induces an spatial displacement, and a re-scaling of the sampled convolutional kernel used during training.

This misalignment is automatically handled under the hood in our `CKConv` implementation. Nevertheless, we highlight this subtle phenomenon to prevent it in future applications.

### E.3 DEALING WITH HIGH-FREQUENCY COMPONENTS

Interestingly, our experiments revealed that our continuous kernels often contain frequency components of frequency higher than the resolution of the sampling grid used during training (Fig. 9). As these high-frequency components are not observed during training, we observe that they hurt performance when evaluated at higher resolutions.

Table 8: Hyperparameter specifications of the best performing CKCNN models.

| PARAMS. | COPY MEMORY | ADDING PROBLEM | sMNIST Small / Big | pMNIST Small / Big | sCIFAR10 Small / Big | CT[†] | SC | SC_RAW[†] | PTB |
|---|---|---|---|---|---|---|---|---|---|
| Epochs | See Appx. E.4 | See Appx. E.4 | 200 | 200 | 200 | 200 | 200 | 300 | 200 |
| Batch Size | 32 | 32 | 64 | 64 | 64 | 32 | 64 | 32 | 24 |
| Optimizer | Adam | Adam | Adam | Adam | Adam | Adam | Adam | Adam | Adam |
| Learning Rate | 5e-4 | 0.001 | 0.001 | 0.001 | 0.001 | 0.001 | 0.001 | 0.001 | 0.002 |
| # Blocks | 2 | 2 | 2 | 2 | 2 | 2 | 2 | 2 | 2 |
| Hidden Size | 10 | 25 | 30 / 100 | 30 / 100 | 30 / 100 | 30 | 30 | 30 | 128 |
| $\omega_0$ | See Appx. E.4 | See Appx. E.4 | 31.09 / 30.5 | 43.46 / 42.16 | 25.67 | 21.45 | 30.90 | 39.45 | 25.78 |
| Dropout | - | - | 0.1 / 0.2 | - | 0.2 / 0.3 | 0.1 | 0.2 | - | - |
| Input Dropout | - | - | 0.1 / 0.2 | 0.1 / 0.2 | 0.0 / 0.0 | - | - | - | 0.1 |
| Embedding Dropout | - | - | - | - | - | - | - | - | 0.1 |
| Weight Dropout | - | - | - | - | - / 0.1 | - | - | - | - |
| Weight Decay | - | - | - | - | - / 1e-4 | - | - | 1e-4 | 1e-6 |
| Scheduler | - | - | Plateau | Plateau | Plateau | Plateau | Plateau | Plateau | Plateau |
| Patience | - | - | 20 | 20 | 20 | 20 | 15 | 20 | 5 |
| Scheduler Decay | - | - | 5 | 5 | 5 | 5 | 5 | 5 | 5 |
| Model Size | 15.52K | 70.59K | 98.29K / 1.03M | 98.29K / 1.03M | 100.04K / 1.04M | 100.67K | 118.24K | 98.29K | 1.8M |

[†] Hyperparameter values for the classification and varying sampling rate tasks. For hyperparameters w.r.t. irregularly-sampled data please see Tab. 9.

In order to neutralize their influence, we filter these components before performing the convolution by means of blurring. This is performed by applying a convolution upon the convolutional kernel with a Gaussian filter $G$ of length $2\frac{\text{sr}_{\text{test}}}{\text{sr}_{\text{train}}} + 1$ and parameters $\mu=0$, $\sigma=0.5$:

$$\left[ G\left(-\frac{\text{sr}_{\text{test}}}{\text{sr}_{\text{train}}}\right), G\left(-\frac{\text{sr}_{\text{test}}}{\text{sr}_{\text{train}}} + 1\right), ..., G(0), ..., G\left(\frac{\text{sr}_{\text{test}}}{\text{sr}_{\text{train}}} - 1\right), G\left(\frac{\text{sr}_{\text{test}}}{\text{sr}_{\text{train}}}\right) \right]$$

Note that blurring is only used when the test sampling rate is higher than the train sampling rate, as opposed to the normalization factor $\frac{\text{sr}_{\text{test}}}{\text{sr}_{\text{train}}}$ discussed in Eq. 5, Appx. A.2, which is applied whenever the sampling rates differ.

### E.4 HYPERPARAMETERS AND EXPERIMENTAL DETAILS

In this section, we provide further specifications of the hyperparameter configurations with with our models are trained. An overview of these hyperparameters is provided in Tab. 8.

**Copy Memory.** We set the number of channels of our CKCNN as to roughly match the number of parameters of the GRU and TCN networks of Bai et al. (2018a). This is obtained with 10 hidden channels at every layer. We observe that the time to convergence grew proportional to the length of the sequence considered. Whereas for sequences of length 100 convergence was shown after as few as 10 epochs, for sequences of length 6000 approximately 250 epochs were required. The maximum number of epochs is set to 50, 50, 100, 200 and 300 for sequences of size 100, 200, 1000, 3000 and 6000. We observe that different values of $\omega_0$ are optimal for different sequence lengths. The optimal $\omega_0$ values found are 19.20, 34.71, 68.69, 43.65 and 69.97 for the corresponding sequence lengths.

**Adding Problem.** We set the number of channels of our CKCNN as to roughly match the number of parameters of the GRU and TCN networks of Bai et al. (2018a). This is obtained with 25 hidden channels at every layer. Similarly to the Copy Memory task, we observe that the time to convergence grew proportional to the length of the sequence considered. Interestingly, this task was much easier to solve for our models, with convergence for sequences of length 6000 observed after 38 epochs. The maximum number of epochs is set to 20, 20, 30, 50 and 50 for sequences of size 100, 200, 1000, 3000 and 6000. We observe that different values of $\omega_0$ are optimal for different sequence lengths. The optimal $\omega_0$ values found are 14.55, 18.19, 2.03, 2.23 and 4.3 for the corresponding sequence lengths.

**sMNIST, pMNIST and sCIFAR10.** We construct two models of different sizes for these datasets: CKCNN and CKCNN-Big. The first is constructed to obtain a parameter count close to 100K. The second model, is constructed to obtain a parameter count close to 1M. The parameters utilized for these datasets are summarized in Tab. 8. Despite our efforts, we observed that our models heavily overfitted sCIFAR10. Combinations of weight decay, dropout and weight dropout were not enough to counteract overfitting.

**CT, SC and SC_raw.** The parameters utilized for classification on these datasets are summarized in Tab. 8. For hyperparameters regarding experiments with irregularly-sampled data please refer to Tab. 9. Any non-specified parameter value in Tab. 9 can be safely consider to be the one listed for corresponding dataset in Tab. 8.

Table 9: Hyperparameter values for experiments on irregularly sampled data. Non-listed parameters correspond to those in Tab. 8.

| PARAMS. | PHYSIONET | CT | | | SC_RAW | | |
|---|---|---|---|---|---|---|---|
| | | (30%) | (50%) | (70%) | (30%) | (50%) | (70%) |
| $\omega_0$ | 4.38 | 17.24 | 12.00 | 4.24 | 35.66 | 31.70 | 25.29 |
| Dropout | 0.0 | 0.2 | 0.2 | 0.0 | 0.1 | 0 | 0 |
| Weight Decay | 0.0 | 0.0 | 1e-4 | 0.0 | 1e-4 | 1e-4 | 1e-4 |
| Batch Size | 1024 | | | | | | |
| Model Size | 175.71K | | 101.75K | | | 99.34K | |

Figure 9: High-frequency components in Sine continuous kernels. We observe that continuous kernels parameterized by Sine networks often contain frequency components of frequency higher than the resolution of the grid used during training. Here, for instance, the kernel looks smooth on the training grid. However, several high-frequency components appear when sampled on a finer grid. Though this may be a problematic phenomenon, we believe that, if tuned properly, these high-frequency components can prove advantageous to model fine details in tasks such as super-resolution and compression cheaply.

**PennTreeBank** For a character-level language modeling on PTB dataset we use hyperparameters specified in Tab. 8. We use embedding of size 100 following the TCN model from Bai et al. (2018a).

