# OpenReview forum: "CKConv: Continuous Kernel Convolution For Sequential Data"
_ICLR.cc/2022/Conference — ICLR 2022 Poster_

### Official Review · Reviewer_Lwkz · 2021-11-01

**Correctness:** 4
**Technical Novelty And Significance:** 3
**Empirical Novelty And Significance:** 2
**Recommendation:** 6
**Confidence:** 3

**Main Review:**

This is a nice paper with one clear idea, which is demonstrated to work well. I like your detailed look at activation functions -- I think you should have a look at "Gegenbauer Polynomials". A potential weakness of the paper is it's limited set of experiments: Compared to the 'baseline' paper Bai 2018a, several datasets are missing, and, given the progression of time, several larger-scale datasets e.g. from speech or the financial domain could be added.

**Summary Of The Paper:**

The paper considers convolutional networks for sequence processing and suggests to parameterize convolutional kernels in this setting by small neural networks. These neural networks are similar to implicit networks in the sense that its input represents a time difference to the currently evaluated time point in a sequence. For each time difference, the associated weights of the convolutional kernel are computed.

This allows to handle arbitrarily large convolutional kernels (albeit the desired size needs to be determined somewhat deterministically apriori), to handle irregularly sampled data and to handle data at different resolutions. Good results on a set of experiments show the practical validity of the idea. The paper does a larger study into the importance of the type of activation functions.



**Summary Of The Review:**

The described idea is a nice transfer from existing work of implicit neural network representations and continuous kernel formulation. Showing the importance of sine activation functions is very helpful to direct attention to the role of activation functions, depending on the actual modeling task. The empirical validation should be larger, though.

---

> ### Author Response · Authors · 2021-11-15
> **First response Reviewer Lwkz**
>
> Dear reviewer Lwkz,
>
> First of all, we would like to thank you very much for your review. We sincerely appreciate the time you spent in evaluating our work, and very much appreciate your comments.
>
> Here we will answer to all of your questions, comments and concerns:
>
> **This allows to handle arbitrarily large convolutional kernels (albeit the desired size needs to be determined somewhat deterministically apriori).** This is true. However, we set our convolutional kernelts to always be as large as the input length. This allows CKConvs to model dependencies across the entire input length.
>
> **I think you should have a look at "Gegenbauer Polynomials"** Thank you very much for your suggestion. We plan to write a thorough comparison among possible ways to parameterize CKConv kernels in the future. These class of polynomials may certainly be of potential interest.
>
> **A potential weakness of the paper is it's limited set of experiments: Compared to the 'baseline' paper Bai 2018a, several datasets are missing, and, given the progression of time, several larger-scale datasets e.g. from speech or the financial domain could be added.** It is true that we do not consider all datasets in Bai 2018a. However, we do include multiple datasets not considered in that paper: sCIFAR10, CharacterTrajectories, SpeechCommands (with Mel features and raw waveforms). In addition, we also provide additional empirical results for settings with missing data and resolution changes, which are also not included in Bai 2018a.
>
> Our dataset selection is chosen as to support each of the claims of our paper. That is, to show that CKConvs not only excel on sequential data, but also on (very long) time-series as well as settings with irregularly sampled data and resolution changes. Based on your comment, however, we have decided to add two additional datasets to our comparison: **PhysioNet** (as in Kidger et al 2020), and  **PennTreebank** (as in Bai et al 2018a). Our results show that CKCNNs obtain a Test AUC score of 0.895 on PhysioNet, outperforming Neural CDEs: the previous state-of-the-art (0.880). For PennTreebank, the experiments are currently being executed, and will be included in our experimental section once they are ready.
>
> Please do let us know if this changes your opinion regarding our empirical evaluation, and if there is anything else we can do to strengthen our paper's positioning with this respect.
>
> **[Final words]** We hope that these responses clarify your questions and concerns. We will reflect this in our manuscript by the end of this week. Please do let us know if you have any follow-up / additional questions.
>
> Best regards,
>
> The Authors

---

> > ### Author Response · Authors · 2021-11-22
> > **State-of-the-art on Char-level PennTreebank**
> >
> > Dear reviewer Lwkz,
> >
> > we have good news with regard to our additional experiments.
> >
> > We are please to inform you that a CKCNN achieves state-of-the-art on Char-level PeenTreebank with a perplexity of **1.045**. In addition, this is obtained with a CKCNN with 1.8M parameters. Note that this uses 60% the number of parameters in a TCN (Bai et al., 2018a), and is 7.4x smaller than TrellisNets (Bai et al., 2018b). Acording to papers with code (https://paperswithcode.com/sota/language-modelling-on-penn-treebank-character), CKCNNs outperform the current state of the art: Mogrifier LSTMs (Melis et al., 2019), while being 13.3x smaller.
> >
> > We hope that this result, combined with the previous results in PhysioNet delivered during this rebuttal period can convince you of the empirical relevance of our approach.
> >
> > **References**
> >
> > Melis, G., Kočiský, T., & Blunsom, P. (2019). Mogrifier lstm. arXiv preprint arXiv:1909.01792.

---

### Official Review · Reviewer_DZHd · 2021-11-02

**Correctness:** 4
**Technical Novelty And Significance:** 3
**Empirical Novelty And Significance:** 3
**Recommendation:** 8
**Confidence:** 3

**Main Review:**

**Pros:**
- Interesting concept of representing arbitrary-sized convolutional kernels.
- The paper is well written, and the method's limitations are outlined clearly
- Great evaluation of how a SIREN-based parametrization is best suited for realizing the MLP.

**Cons:**
- No code is provided during the review (although the authors claim to release it)
- The experimental evaluation mostly consists of equidistantly sampled fixed-length time series (sMNIST, etc.), i.e., data for which CKConv provides only little benefits over standard CNNs.
- Experimental evaluation with attention-based architectures on irregularly sampled variable-length sequences datasets, e.g., PhysioNet, MIMIC-III, and the Human Activity dataset, from Shukla et al. 2021 would have demonstrated the advantages and limitations much better.

**Questions**:
- How was the sequence reduced to a single prediction in the sequence classification tasks (sMNIST, etc.)?
- The paper Gu et al. 2020 "HiPPO: Recurrent Memory with Optimal Polynomial Projections" was published at NeurIPS. Please update the reference.

Shukla et al. Multi-Time Attention Networks for Irregularly Sampled Time Series. ICLR 2021

**Summary Of The Paper:**

The paper proposes continuous convolutional kernels parametrized in the form of an MLP. The MLP gets the relative time as input and outputs the column of the convolutional kernel at the given relative time. The authors show that Sine-based (SIREN) non-linearity is best suited for the kernel generating MLP, and an experimental evaluation is performed.

**Summary Of The Review:**

I am in favor of acceptance as the concept of CKConv provides enough contribution, although the experimental evaluation would have been stronger with standard irregularly time-series datasets (PhysioNet, MIMIC-III)

---

> ### Author Response · Authors · 2021-11-15
> **First response Reviewer DZHd**
>
> Dear reviewer DZHd,
>
> First of all, we would like to thank you very much for your review. We sincerely appreciate the time you spent in evaluating our work, and very much appreciate your comments.
>
> Here we will answer to all of your questions, comments and concerns:
>
> **No code is provided during the review (although the authors claim to release it)** Our code is actually already online in a non-anonymous repository. This is why we did not include the link in our paper. To allow you access to our code, we will provide an anonymized copy of our code as supplementary material to our submission. Aside from the source code, our repository contains all pretrained models, as well as the precise commands required to replicate our experiments. In addition, we provide an small demo illustrating how to use CKConvs for other projects, and slides outlining our work.
>
> **The experimental evaluation mostly consists of equidistantly sampled fixed-length time series (sMNIST, etc.), i.e., data for which CKConv provides only little benefits over standard CNNs.**  Based on your comment, we have now included the PhysioNet (as in Kidger et al 2020) in our experiments. Our results show that CKCNNs obtain a Test AUC score of 0.895 on PhysioNet, outperforming Neural CDEs: the previous state-of-the-art (0.880).
>
> Given the constrained time during this rebuttal period, we are unable to perform experiments on the MIMIC-III, and the Human Activity dataset too. However, if you find it necessary to include these datasets in our comparison, we will make sure to do so for the camera-ready version of our work.
>
> **Experimental evaluation with attention-based architectures on irregularly sampled variable-length sequences datasets, e.g., PhysioNet, MIMIC-III, and the Human Activity dataset, from Shukla et al. 2021 would have demonstrated the advantages and limitations much better.** We now now included a Transformer model in our comparisons for time-series. In particular, we observe that a transformer as in Trinh et al. 2018 obtains 90.75% accuracy on SC, underperforming the 95.3 accuracy of CKConv. In addition, due to the quadratic complexity of self-attention, this transformer is unable to operate on the raw SpeechCommands dataset (sequence length = 16000). CKConvs, on the other hand, do not present quadratic complexity, and thus, are able to get good performance in this difficult dataset.
>
> We will include a word in our discussion regarding transformer architectures, and will compare to Shukla et al. 2021 in our experiments on irregularly-sampled data.
>
> **How was the sequence reduced to a single prediction in the sequence classification tasks (sMNIST, etc.)?** The images are flattened and presented to the sequential model as a sequence of pixels. The model must classify the digit based on the representation at the last pixel's position.
>
> **The paper Gu et al. 2020 "HiPPO: Recurrent Memory with Optimal Polynomial Projections" was published at NeurIPS. Please update the reference.** Thank you for pointing this out.
>
> **[Final words]** We hope that these responses clarify your questions and concerns. We will reflect this in our manuscript by the end of this week. Please do let us know if you have any follow-up / additional questions.
>
> Best regards,
>
> The Authors

---

> > ### Comment · Reviewer_DZHd · 2021-11-22
> > **I increased my score**
> >
> > I increased my score because I am quite satisfied with the provided answers and experiments

---

### Official Review · Reviewer_NgYq · 2021-11-03

**Correctness:** 4
**Technical Novelty And Significance:** 3
**Empirical Novelty And Significance:** 3
**Recommendation:** 8
**Confidence:** 4

**Main Review:**

The contribution is clear and well written. I reviewed a previous
version for another venue and thank the authors for their effort in
clarifying their manuscript, which I now find very didactic and
pleasant to read.

I found the concept of continuous kernel interesting. The authors make
it clear that they are adapting it from previous work on 3D data, but
also show how its transfer to sequential data is a useful contribution
in particular for dealing with long-range dependencies without the
limitations of RNN or discrete CNN. In particular, they clearly show
how a discrete convolution kernel would require much more parameters
than a CKConv to model the same range of dependencies.

They also provide thorough experiments illustrating the superiority of
their proposed continuous kernel CNN (CKCNN) against state of the art
approaches on all targeted applications: long-range dependency
modelling, irregular sampling, varying sampling rate.

I believed that this contribution could both be inspiring for new
paradigms of networks (e.g. as an alternative or complement to
attention), and lead to better performances in practice on the
targeted applications.

One point that could be further clarified is the choice of using a single MLP
for modelling the Nin x Nout convolution kernels. Among other
possibilities, a separate network could be used for each of the Nout
convolution filters (i.e., one MLP instead of one discrete Nin x
kernel_size filter). Using a single MLP likely creates dependencies
among the Nout filters, is there a justification for this choice?


**Summary Of The Paper:**

This contribution adapts to sequential data the paradigm of continuous
convolution kernels in convolutional neural networks (CNNs). While
usual kernels are an extensive list of weights (one for each
position), the introduced CKConv frames these kernels as continuous 1D
functions, parameterized by a small multilayer perceptron. This
paradigms allows for wider kernels with fewer parameters, making it
possible to deal with long-range dependencies without
recurrence. Several experiments are provided and show that the
resulting CNN achieves state of the art performances on a variety of
tasks, and in particular is able to deal with irregularly sampled
data.


**Summary Of The Review:**

The contribution is original and could have a large impact both because it leads to better performances in several important situations (including long range dependencies) and because it could inspire new designs. The proposed method is thoroughly evaluated.
The submission is also clear and well written.

---

> ### Author Response · Authors · 2021-11-15
> **First response Reviewer NgYq**
>
> Dear reviewer NgYq,
>
> First of all, we would like to thank you very much for your review., and we are very glad to hear that you like the improvements we have made to our work. We sincerely appreciate the time you spend in evaluating our work.
>
> **One point that could be further clarified is the choice of using a single MLP for modelling the Nin x Nout convolution kernels. Among other possibilities, a separate network could be used for each of the Nout convolution filters (i.e., one MLP instead of one discrete Nin x kernel_size filter). Using a single MLP likely creates dependencies among the Nout filters, is there a justification for this choice?** This is a very good point. We will clarify this in an updated version of our manuscript.
>
> Based on the empirical observation that Sine networks were able to fit very complex functions with little overall dependencies (e.g., a sequence of random noise), we decided to use a single network that predicts Nout x Nin channels in order to reduce the number of parameters required for the parameterization. In addition. we realized that hidden representations at  a certain layer could be re-used to model several different functions (just as a basis of random Sine / Cosine functions can be combined with different weights to model several different functions). Consequently, we figured that it is much more parameter efficient (and perhaps even easier from an optimization perspective) to learn a single shared hidden representation, and use different weights to generate Nout x Nin different kernels.
>
> We evaluated this hypothesis by fitting a number of images from the CIFAR10 dataset simultaneously (i.e., treating different images as extra output channels) with a single MLP. We observed that the MLP was able to reconstruct these images using a single shared hidden representation, although the output channels were not strongly correlated.
>
> We will include a discussion regarding this point to the main text of our work, and will add these additional to the appendix.
>
>  **[Final words]** Please do let us know if you have any follow-up / additional questions.
>
> Best regards,
>
> The Authors

---

### Official Review · Reviewer_nLYG · 2021-11-04

**Correctness:** 4
**Technical Novelty And Significance:** 3
**Empirical Novelty And Significance:** 3
**Recommendation:** 8
**Confidence:** 4

**Main Review:**

# Review
## Clarity and novelty
This is overall a well-written paper, which clearly discusses the limitations of popular sequence models such as CNNs and RNNs, and explains how the proposed CKConv model is able to provide improvements in those aspects. The related work seems to be adequately cited. The main idea, which is to represent the convolution kernel implicitly via parametrizing it by a neural network, is not novel in the literature, but it has not been applied previously for sequential data modelling as far as I know.
***
## Strengths
The main value of the paper for me is in:
1) adapting the continuous kernel convolution model to the sequential data context
   + the idea of using a CKConv with a global memory horizon seems particularly useful, remedying the problem of not being able to learn long-range relationships, typically associated with RNNs and fixed horizon (shallow) CNNs
   + the ability to generate kernel weights for arbitrary relative offsets naturally also lends itself to learning from irregularly sampled data, which is non-trivial to address with classic approaches
2) the effort put into the empirical evaluation
   + observing that convolution kernels might naturally contain high-frequency oscillations, which are difficult to learn via common activations, and identifying MLPs with Sine activations as a suitable candidate for representing the kernel
   + carrying out the empirical evaluation across a range of tasks, which show strong performance of the proposed model
***
## Drawbacks
My criticism is regarding some of the claims and conclusions drawn. See these points below.

**Insensitivity to resolution changes**
It seems that the relation in equation (5) does not exactly support that the model is robust with respect to larger resolution changes. This is also shown in Table 5, where the performance degrades significantly if there is an order of magnitude difference between training and testing resolutions. This is of course because the discretized convolution in eq.(2) is not a proper discretization of the integral in eq.(1). In particular, if we consider a high-frequency limit (that is, sampling a continuous signal at increasing frequencies), then the summation in (2) does not converge. I wonder why the authors chose to omit the normalization from eq.(2)? They could have normalized by the number samples (or equivalently, by the sampling rate) for equispaced samples, or for irregular data simply multiply the summand by the local step size to obtain Riemann sums that are actually convergent, and hence mapping that is continuous-time convergent and invariant to resolution changes, which it is not right now.

**Asynchronous sampling**
It seems to me that regarding asynchronously sampled data, the model formulation is equivalent to combining zero imputation for the missing values with concatenating an observation mask to the input as extra channels. This is a valid approach, but not specific to the model at hand.

**Activations in $\texttt{MLP}^\psi$**
I found the discussion regarding the activation functions in the kernel parametrization quite intriguing. In particular, the experiments shown in Figure 4 coincide with the findings of "On the Spectral Bias of Neural Networks", Rahaman et al. 2019, where it is observed that NNs with ReLU activations trained under GD are biased towards learning low-frequency functions. It is also an interesting finding that the Sine activation seems to be able to overcome this limitation in this setting. But then one of the main points the paper tries to communicate seems to be that _Neural networks parameterizing spatial functions should use Sine nonlinearities_, which in my opinion paints too dark of a picture about ReLU and its companions. Especially so, as there have been approaches proposed in the literature to circumvent this limitation without changing the activation itself, see "Fourier Features Let Networks Learn
High Frequency Functions in Low Dimensional Domains", Tancik et al. 2020. Without properly testing out if these techniques do help the usual activations, the aforementioned conclusion stands on shaky grounds. Especially so, since ReLUs are well-studied in the literature, while Sine nonlinearities in NNs (i.e. SIRENs) seem to be somewhat of a wildcard. In particular, by the discussion in App. E.3 I am left wondering if SIRENs might be, on the other hand, prone towards learning (unnecessarily) high-frequency functions?

Concretely, I would be curious to know how the conclusions of the experiment in Figure 4, and the ablation study in App. D.1 change if the relative time offset $\Delta \tau$ is first passed through a positional encoding or Random Fourier Feature mapping as done in the previously referenced paper. Furthermore, in case any of these preprocessings do help the "standard" activations, it would be also good to know if the learned functions still contain the high-frequency oscillations beyond the input time grid as mentioned in App. E.3, or if these end up being smoother?

**Modelling long-range relationships**
The authors have noted that dilated convolutions are able to model long-range interactions. I wonder why it wasn't included among the experiments? Clearly, to achieve a larger receptive field, dilated convolutions stack many layers, which means that the inputs are passed through several nonlinear activations before reaching the final representation the prediction layer receives. On the other hand, the CKConvs can increase the receptive field while keeping the model shallow. The benefits of this are not really clear at the moment, and whether the way the sequential information is processed in the dilated model (by passing it through "extra depth") has adversarial effects for the quality of the results.
Additionally, self-attention models are also able to capture long-range dependencies in their input sequences, so it would be worthwhile to include a discussion about them, and compare against it on more experiments.

**Summary Of The Paper:**

The paper introduces convolutions with continuously parametrized kernels for sequential data. Continuous kernel convolutions, CKConv for short, parametrize the kernel associated to a convolutional layer as a continuous mapping, $\psi: \mathbb{R}^+ \rightarrow \mathbb{R}^{N_{out} \times N_{in}}$, from the relative time offset $\Delta\tau \in \mathbb{R}^+$ to a weights matrix $\mathbf{W} \in \mathbb{R}^{N_{out} \times N_{in}}$, i.e. $\psi(\Delta \tau) = \mathbf{W}$. This is in contrast to traditional CNN approaches, which represent the convolution kernel as an explicitly learnable set of weights over an a-priori fixed receptive field.

The corresponding CKConv operation can be applied
1) over arbitrarily long receptive fields, including a global one over the history of the input sequence, without the heavy memory burden incurred by representing the convolution kernel explicitly as a set of weights along the horizon;
2) to sequences that are sampled irregularly or asynchronously, since the admissible set of relative positions are not fixed in advance.

A relation is given, which shows approximately how the CKConv changes under resampling the input sequence (the output gets rescaled by the ratio of the sampling rates), which is suggested by the authors to support well-behavedness under varying input resolutions during training and/or between training and testing. Afterwards, a discussion is given about the parametrization of the kernel. Specifically, various MLP models are considered with different activation functions, such as, ReLU, LeakyReLU and Swish, and it is demonstrated that these MLPs are unable to learn functions with high-frequency oscillations, while a recently introduced Sine activation is able to.

The experimental evaluation includes 1) classic stress tests for RNNs, 2) discrete sequence tasks (sMNIST, pMNIST, sCIFAR10), 3) time series modelling (CharacterTrajectories, Speech Commands) , where additionally robustness to missing/irregularly-sampled data and resolution changes are also investigated. The proposed model overall performs favourably to common alternatives; and it seems robust with respect to missing data, and moderate resolution changes.

**Summary Of The Review:**

I am in favour of the paper mainly due to the compelling empirical performance, which should be of interest to the community, although some additional baselines and experiments could have been included.  It is clearly written with figures and illustrations to aid the reader. The main idea is not novel, but it has not been adapted to handling sequential data before, and it seemingly has many benefits in this context.  My criticism is regarding some discussions in the paper, and that some properties of the model seem to be overstated.

POST-REBUTTAL: Given the comprehensive answers given by the authors, and the implemented changes in the paper, I am inclined to raise my rating to an 8 from the previous 6.

---

> ### Author Response · Authors · 2021-11-15
> **First response Reviewer nLYG**
>
> Dear reviewer nLYG,
>
> First of all, we would like to thank you very much for your thorough and insightful review. We sincerely appreciate the time you spent in evaluating our work, and very much appreciate your comments.
>
> Here we will answer to all of your questions, comments and concerns:
>
> **[Summary of the paper]**:
>
>  **The corresponding CKConv operation can be applied over arbitrarily long receptive fields, including a global one over the history of the input sequence, without the heavy memory burden incurred by representing the convolution kernel explicitly as a set of weights along the horizon.** Additionally, we require much less parameters in order to model convolutional kernels of a given length. This results in small networks with large convolutional kernels, which positively influences the statistical efficiency of our method.
>
> **[Drawbacks]**:
>
> **Insensitivity to resolution changes** It is true that our method is not exactly robust to resolution changes. However, to the best of our understanding, the reason of this behavior is different to your interpretation. In particular, we believe this error appears as an effect of aliasing. Since SIRENs often contain frequency components at frequencies higher than the resolution used during training, e.g.,Fig. 9, sampling a SIREN at a higher resolution will not be band-limited by the resolution on which training happened and thus, the function that the SIREN represents at this resolution will be different from that the SIREN was trying to model at the original resolution. We believe that is precisely this effect what leads to the inconsistency across resolutions. Although we alleviate this effect by using low-pass filtering (Appx. E.3), this is not sufficient to solve the problem entirely.
>
> In subsequent work we have found a different parameterization of the convolutional kernel, with which the maximum frequencies of a convolutional kernel can be controlled analytically. This allows CKConvs to create kernels that are bandlimited regardless of the sampling rate. This allows us to create CKConv kernels that extrapolate better to unseen resolutions. With that being said, we emphasize that, despite these imperfections, CKConvs generalize much better across resolutions than existing methods.
>
> We will clarify this is an updated version of our manuscript.
>
> **Asynchronous sampling** This is true. We rely on this imputation method to be able to used vectorized primitives in deep learning frameworks. However, it is not necessary for our method to work. For instance, we could implement a convolution operation, i.e., unfold + matrix multiplication, that receives functions and coordinates instead of functions on a grid, and use this implementation for the computation of the convolution. Nevertheless, existing methods that rely on this strategy are much slower than simply using the aforementioned vectorized primitives, e.g., https://github.com/mfinzi/LieConv/blob/master/lie_conv/lieConv.py#L72 .
>
> In addition, we would like to emphasize that this imputation method is also used by other methods that claim to be able to handle irregularly-sampled data, and are specifically designed for that purpose, e.g., Kidgerr et al. 2020 ( https://github.com/patrick-kidger/NeuralCDE/blob/7e529f58441d719d2ce85f56bdee3208a90d5132/experiments/datasets/common.py#L63 ).

---

> > ### Author Response · Authors · 2021-11-15
> > **First response Reviewer nLYG -- continuation --**
> >
> > **Activations in MLP$^\psi$**. You raise a very important point here. You are completely right. We should have not stated that *neural networks parameterizing spatial functions should use Sine nonlinearities*, but that they should use parameterizations able to model high-frequencies, instead. We will adjust this in our manuscript. We use SIRENs in this work since early studies illustrated that SIRENs were very well suited to model functions with high-frequencies. However, we did not consider Random Fourier Networks (Tancik et. al. 2020) in these early studies.
> >
> > To accomodate for this, we will adjust our claim as mentioned in the previous paragraph. In addition, we have followed your suggestion, and replicated the experiment in Figure 4 using Random Fourier Networks (RFNets). Intiguingly, we found that RFNets are able to get better approximations that SIRENs in some cases. Based on these interesting findings, we will include these results in the paper, and accentuate the importance of finding optimal paramterizations for the kernels in CKConvs, not restricted to SIRENs.
> >
> > Additionally, we will include RFNets in the ablation studies of Appx. D.1. If you find it adequate, we could also provide results for all of our experiments with RFNet kernels as well. However, based on the limited time we have for this rebuttal, we cannot promise to have all of these experiments ready this week. We can promise however, to add them to the final version of our manuscript.
> >
> > Regarding the high-frequency oscillations, we have observed that RFNets still suffer from this behavior. In particular, the larger the Gaussian scale of $\mathbf{B}$ in $\gamma$ upon initialization, the larger these oscillations become.
> >
> > **Modelling long-rage relationships** We do consider baselines with dilated convolutions in our comparison. The Temporal Convolutional Networks of Bai et al. 2018a are CNNs with dilated convolutional kernels very much like WaveNets (van der Oord, 2016).
> >
> > In addition, based on your suggestion, we have now included a Transformer model in our comparisons for time-series. In particular, we observe that a transformer as in Trinh et al. 2018 obtains 90.75% accuracy on SC, underperforming the 95.3 accuracy of CKConv. In addition, due to the quadratic complexity of self-attention, this transformer is unable to operate on the raw SpeechCommands dataset (sequence length = 16000). CKConvs, on the other hand, do not present quadratic complexity, and thus, are able to get good performance in this difficult dataset. In addition to these results, we will include a word in our discussion regarding transformer architectures.
> >
> > **[Final words]** We hope that these responses clarify your questions and concerns. We will reflect this in our manuscript by the end of this week. Please do let us know if you have any follow-up / additional questions.
> >
> > Best regards,
> >
> > The Authors

---

> > > ### Author Response · Authors · 2021-11-23
> > > **Regarding Random Fourier Features**
> > >
> > > Dear reviewer nLYG,
> > >
> > > we have some news regarding the comparison with Random Fourier Networks (RFNets) (Tancik et al. 2020).
> > >
> > > We have compared  RFNets to SIRENs in the experiments in Fig 4 and the ablation studies in Appx. D.1. Interestingly, we have found that RFNets outperform SIRENs in several of these tasks! This is a very interesting finding and we thank you very much for that observation.
> > >
> > > Although we could include these results in the paper, we think that these observations are actually worth a paper of their own. Subsequent work has come across other kernel parameterizations that also outperform SIRENs as kernel parameterization (link not provided here for the sake of the double-blind review). We therefore believe that a thorough and comprehensive empirical (and perhaps theoretical) evaluation of possible kernel parameterizations is worth exploring. In order not to hamper the contribution of such a work, we are inclined to think that it is better not to include these experiments to the paper, but rather wait for a more extensive comparison of multiple approaches in the future.
> > >
> > > With that being said, we are more than happy to hear your comments. If you think that we should nevertheless include this comparison in our paper, we will include it. If this is the case, however, we would like to include it as part of our camera-ready version. This would give us enough time to run experiments with RFNets across all of the datasets considered and provide a better comparison to SIREN kernels.
> > >
> > > **Neural networks parameterizing spatial functions should be able to model high-frequencies.** Based on your previous observations as well as the new results obtained with RFNets, we have now changed the conclusion in our Discussion section stating that works should transition towards Sine nonlinearities by the more accurate statement that *Neural networks parameterizing spatial functions should be able to model high-frequencies.*.
> > >
> > > In addition, we have included Tancik et al., 2020 as one of the possible alternative kernel parameterizations in our Future Work section.
> > >
> > > Please let us know what you think.
> > >
> > > Best regards,
> > >
> > > The Authors.

---

> > > > ### Comment · Reviewer_nLYG · 2021-11-27
> > > > **Thank you for your comprehensive answers**
> > > >
> > > > I would like to thank the authors for the comprehensive answers to my points, which is adapting the discussion in the paper based on the feedback, and doing additional experiments: including the transformer baseline, and carrying out the requested experiments with kernels parametrized by RFNets. I am satisfied with the answers given by the authors to my questions. In particular, I am pleased that the suggested Fourier Feature Networks turned out to be a fruitful direction for improvements.
> > > >
> > > > As the performance of the model is already strong, I do not think further improvements are necessary for publication. I feel that if the the RFNets were to be included and they do end up outperforming SIRENS, this would require too many post-rebuttal changes in the discussion. I also see that the authors have indicated using the RFNets of Tancik et al. as potential future work, and updated their discussion on kernel parametrizations. There are still questions left and follow up work to be carried out, but these seem to be clearly communicated, and the contributions are quite sufficient for publication as is.  With these points in mind, I am raising my rating to an 8.

---

### Decision · Program_Chairs · 2022-01-20

**Decision:**

Accept (Poster)

**Comment:**

This paper introduces a convolution where the kernel is parametrised continuously over time (in the context of recurrent networks) to address vanishing gradients issues, by using another neural network to generate the kernels.
This is a meaningful idea, addressing an important problem.
The paper is well written and clear. The idea is novel (parametrised kernel already exist, but the way it's used here is new).
The experimental section is solid, although some reviewers suggests it could be extended with more baselines.
All reviewers recommend to accept the paper, therefore I also recommend accept.